JCB Journal of Cell Biology

# The motor domain of the kinesin Kip2 promotes microtubule polymerization at microtubule tips

Xiuzhen Chen[1], Didier Portran[2]*, Lukas A. Widmer[3]*, Marcel M. Stangier[4]*, Mateusz P. Czub[4], Dimitris Liakopoulos[2,6], Jörg Stelling[3], Michel O. Steinmetz[4,5], and Yves Barral[1]

**Kinesins are microtubule-dependent motor proteins, some of which moonlight as microtubule polymerases, such as the yeast protein Kip2. Here, we show that the CLIP-170 ortholog Bik1 stabilizes Kip2 at microtubule ends where the motor domain of Kip2 promotes microtubule polymerization. Live-cell imaging and mathematical estimation of Kip2 dynamics reveal that disrupting the Kip2–Bik1 interaction aborts Kip2 dwelling at microtubule ends and abrogates its microtubule polymerization activity. Structural modeling and biochemical experiments identify a patch of positively charged residues that enables the motor domain to bind free tubulin dimers alternatively to the microtubule shaft. Neutralizing this patch abolished the ability of Kip2 to promote microtubule growth both in vivo and in vitro without affecting its ability to walk along microtubules. Our studies suggest that Kip2 utilizes Bik1 as a cofactor to track microtubule tips, where its motor domain then recruits free tubulin and catalyzes microtubule assembly.**

## Introduction

Over the last decade, several kinesin-related motor proteins emerged as key players in controlling microtubule dynamics in a wide range of cellular processes, including cell division, cargo transport, organelle distribution, and cell polarity (reviewed in Lu and Gelfand, 2017; Brouhard and Rice, 2014; Drummond, 2011). Non-motile kinesins from the kinesin-13 family were first identified as microtubule depolymerases and shown to promote microtubule catastrophes, i.e., the switch from growth to shrinkage (Walczak et al., 1996; Hunter et al., 2003). Subsequently, motile kinesins such as kinesin-8 family members (budding yeast Kip3, fission yeast Klp5/6, human Kif18a, and Kif18b) were also found to promote microtubule depolymerization (Gupta et al., 2006; Varga et al., 2006; Mayr et al., 2007; Tischer et al., 2009). Extensive studies using in vitro reconstitution of purified components or acquiring structural details of their motor domains in complex with tubulin have greatly advanced our understanding of how these motor proteins possess both motility and microtubule depolymerase activities (Arellano-Santoyo et al., 2017; Varga et al., 2006). An emerging theme is that the motor domain of these kinesins binds preferentially to the "curved" conformational state of the αβ-tubulin heterodimer at microtubule plus-ends over its "straight" conformation found in microtubule lattices (Brouhard and Rice, 2014). Thereby, they stabilize curved protofilaments at plus-ends and facilitate microtubule catastrophe (Brouhard and Rice, 2014).

Interestingly, other kinesins promote microtubule growth instead, including the kinesin-5 Eg5, the kinesin-7 CENP-E, and the budding yeast kinesin Kip2 (Hibbel et al., 2015; Chen and Hancock, 2015; Sardar et al., 2010). Eg5 and CENP-E promote microtubule elongation potentially through stabilizing newly incorporated tubulin dimers at the microtubule plus-end (Sardar et al., 2010; Chen et al., 2019a; Chen and Hancock, 2015). Consistent with the ability of the cytoplasmic kinesin Kip2 to stabilize microtubules, deletion of the *KIP2* gene shortens astral microtubules that become much less abundant throughout the cell cycle (Huyett et al., 1998; Carvalho et al., 2004; Cottingham and Hoyt, 1997). Conversely, Kip2 overexpressing cells form long astral microtubules (Carvalho et al., 2004). However, how Kip2 promotes microtubule growth or stabilizes microtubules is not clear. At the molecular level, Kip2 is composed of a disordered N-terminal domain (residues 1–99) that can be phosphorylated (Drechsler et al., 2015), followed by a microtubule-dependent motor domain (residues 100–503) and a C-terminal, coiled-coil

[1]Institute of Biochemistry, Eidgenössische Technische Hochschule Zürich, Zurich, Switzerland; [2]CRBM, Université de Montpellier, CNRS, Montpellier, France; [3]Department of Biosystems Science and Engineering, Eidgenössische Technische Hochschule Zürich, and Swiss Institute of Bioinformatics, Basel, Switzerland; [4]Department of Biology and Chemistry, Laboratory of Biomolecular Research, Paul Scherrer Institute, Villigen, Switzerland; [5]University of Basel, Biozentrum, Basel, Switzerland; [6]Laboratory of Biology, University of Ioannina, Faculty of Medicine, Ioannina, Greece.

*D. Portran, L.A. Widmer, and M.M. Stangier contributed equally to this paper.   Correspondence to Yves Barral: yves.barral@bc.biol.ethz.ch

X. Chen's current affiliation is Cancer Biology and Genetics Program, Memorial Sloan Kettering Cancer Center, New York, NY, USA.

dimerization domain (residues 504–706; Fig. 1, A and B). In vitro, Kip2 is a highly processive motor that accelerates microtubule growth and inhibits catastrophes without cofactor (Hibbel et al., 2015; Roberts et al., 2014). Remarkably, the microtubule growth promoting activity of Kip2 in vitro is at least 20 times stronger than that of Eg5 or CENP-E under similar conditions (Hibbel et al., 2015; Chen and Hancock, 2015; Sardar et al., 2010). In contrast, in vivo Kip2 cannot stabilize microtubules in the absence of the cytoplasmic linker protein (CLIP) Bik1, the *Saccharomyces cerevisiae* ortholog of metazoans' CLIP-170. Loss of either Bik1 or Kip2 results in short cytoplasmic microtubules, and the effects of these mutations are not additive, indicating that both proteins require each other for proper function (Berlin et al., 1990; Caudron et al., 2008; Cottingham and Hoyt, 1997; Miller et al., 1998; Huyett et al., 1998). Consistent with this interpretation, overexpression of Kip2 results in the hyper-elongation of cytoplasmic microtubules only in the presence of Bik1 (Carvalho et al., 2004). However, the relative contributions of Kip2 and Bik1 in this process are unclear. While it has been proposed that Kip2 acts on microtubules simply by transporting Bik1 toward the plus-end of microtubules and enriching it there (Carvalho et al., 2004), this model does not account for the fact that Kip2 can promote microtubule growth on its own in vitro.

The CLIP proteins form a family of highly conserved proteins that localize to and track microtubule plus-ends (+TIP proteins) in Eukaryotes (reviewed in Komarova et al., 2002). There, they mediate interactions between microtubule plus-ends and a broad range of cellular structures such as endosomes (Pierre et al., 1992) and kinetochores (Tanenbaum et al., 2006). They also promote microtubule growth as demonstrated for CLIP-170 in mammalian cells (Komarova et al., 2002), Tip1 in fission yeast (Brunner and Nurse, 2000), and Bik1 in budding yeast (Carvalho et al., 2004). CLIP orthologs are generically composed of an N-terminal microtubule-binding region, a long central coiled-coil that enables dimerization, one (Bik1) or two (CLIP-170) zinc-knuckle domains, and a C-terminal EEY/F linear sequence motif. The N-terminal microtubule-binding region contains one (Bik1) or two (CLIP-170) cytoskeleton-associated protein glycine-rich (CAP-Gly) domains, which recognize C-terminal EEY/F motifs on proteins such as α-tubulins, end-binding proteins (EBs), and on CLIP proteins themselves (Akhmanova and Steinmetz, 2008). These proteins track growing microtubule plus-ends indirectly through binding EBs, even though they can bind tubulin directly (Dixit et al., 2009; Bieling et al., 2008). Their C-terminal zinc-knuckle domains mediate association with the minus-end directed motor protein dynein through the adaptor protein LIS1 to control the spatial distribution of dynein (Kardon and Vale, 2009; Coquelle et al., 2002). In mammalian cells and budding yeast, CLIP-170 and Bik1, respectively, use their N-terminal microtubule-binding region to stimulate microtubule growth (Arnal et al., 2004). However, it is unknown whether they promote microtubule growth directly—through binding and somehow stabilizing or promoting tubulin incorporation—or indirectly through binding and recruiting effectors.

Here, we employ cell biology, biophysical approaches, and mathematical modeling to address the following questions: First, how do Bik1 and the motile kinesin Kip2 interact with each other and facilitate each other's function in vivo? Second, how does Kip2 exert robust microtubule polymerase activity in vivo, and how is this activity coordinated with its motility?

## Results

### Kip2 promotes microtubule polymerization in vivo

The ability of Kip2 to promote microtubule growth in vivo is well documented (Caudron et al., 2008; Cottingham and Hoyt, 1997; Huyett et al., 1998). However, it is still to be determined whether it does so merely by inhibiting microtubule catastrophe or by promoting microtubule polymerization. Addressing this question requires measuring the dynamic instability parameters of microtubules with sufficient precision within the 3D volume of the cell, i.e., requires tracking the microtubules moving in and out of any given focal plane. Through optimizing our microscopy setup, we succeeded in tracking yeast astral microtubules throughout the cell volume at nearly 1 s time resolution, which was about four times faster than previous studies (Fees et al., 2017). The yeast astral microtubules' 3D length was measured in time-lapse movies of preanaphase cells using Bik1-3xGFP and Spc72-GFP as microtubule plus- and minus-end markers, respectively (Stangier et al., 2018). This allowed us to track each microtubule as two particles instead of a rod, which greatly improved the performance of image processing and allowed us to reduce exposure time to 30 ms per plane (Fig. 1, C and D; Stangier et al., 2018). To compute the maximum length and the lifetime of each recorded cytoplasmic microtubule, 3D coordinates of both the plus- and the minus-end of each microtubule were extracted over all time frames (17 z-sections 0.24 μm apart, one full volume every 1.07 s). As a baseline, only 15% (302 out of 2,001) of the *kip2Δ* mutant cells formed detectable astral microtubules (longer than 667 nm, owing to the microscope resolution) over the imaging window of 85.6 s (80 frames), whereas 100% (511 out of 511) of the wild-type cells did so (Fig. 1, C and F; and Fig. S1, A and B; all lengths of microtubules below the detection limit are noted as 0 μm). Furthermore, the detectable microtubules of *kip2Δ* mutant cells were significantly shorter (1.4 ± 0.02 μm, mean ± SD, n = 3 independent clones, P < 0.0001, ordinary one-way ANOVA test) than those of wild-type cells (2.1 ± 0.1 μm, mean ± SD, n = 3 independent clones; Fig. 1, G and H; and Table S1). Loss of Kip2 increased the catastrophe frequency nearly twofold and reduced microtubule growth speed by 25% (Fig. 1 I and Table S1). Together, these data suggest that Kip2 not only stabilizes but also promotes microtubule growth in vivo.

### The C-terminal tail and the motor domain of Kip2 are indispensable for astral microtubule growth in vivo

To identify the domains of Kip2 that are required for polymerizing microtubules in vivo, we generated several truncation alleles expressed from the endogenous locus (Fig. 1 A) and determined how they affected microtubule growth. Kip2-ΔN lacks the N-terminal disordered region (residues 1–99) that is enriched in phosphorylated serines and threonines (Drechsler

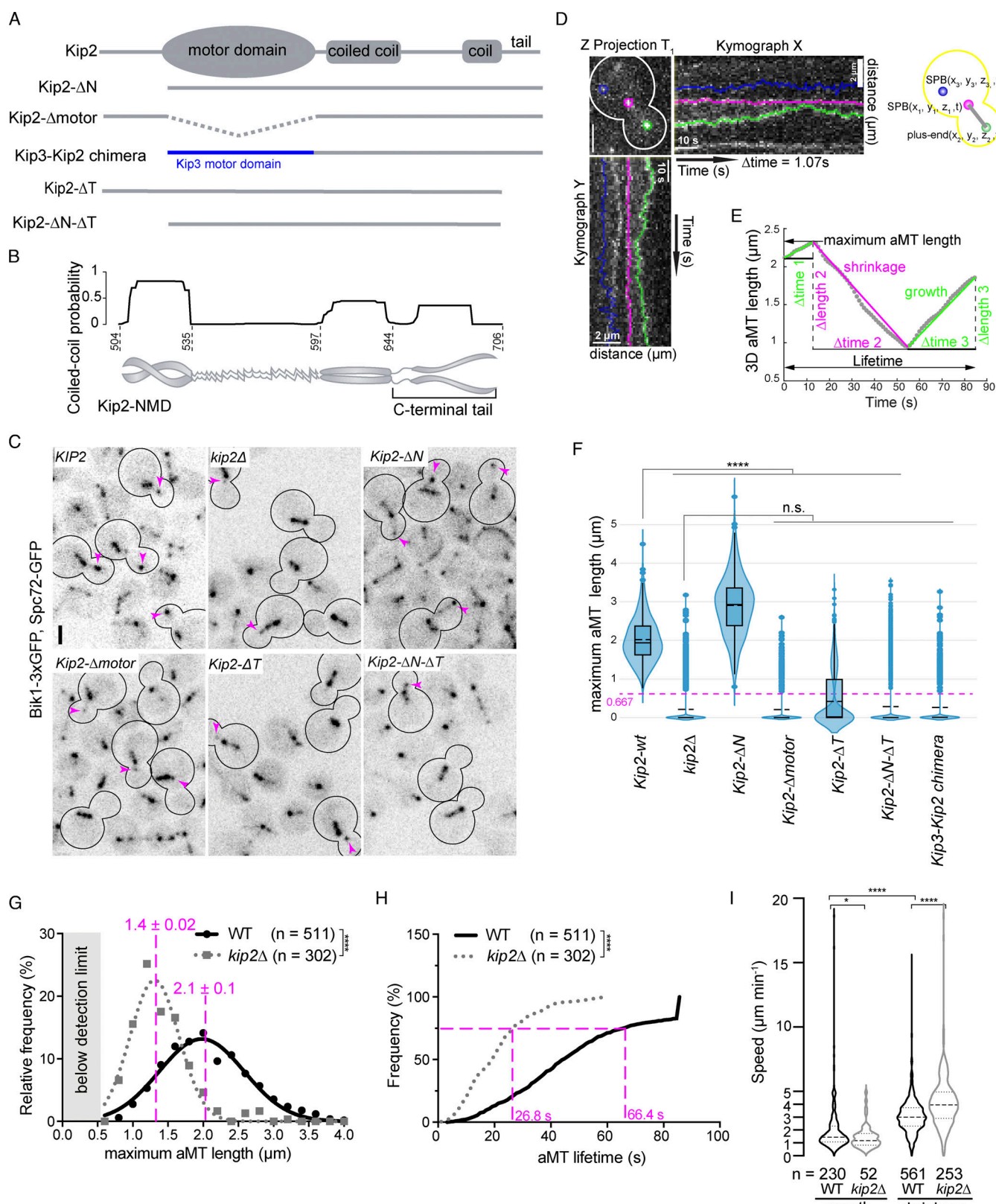

Figure 1. **Kip2 motor domain and the C-terminal tail domain are essential for microtubule growth in vivo. (A)** Schematic representations of protein constructs used. **(B)** Analysis of the Kip2-NMD amino acid sequence using the MultiCoil-prediction server with a cutoff of 0.5. **(C)** Representative images of preanaphase cells of indicated genotype expressing Bik1-3xGFP and Spc72-GFP. Scale bars, 2 µm. Magenta arrowheads mark the plus-ends of cytoplasmic microtubules. **(D)** The maximum intensity z-projection of time point one of a representative preanaphase cell. The 3D coordinates of aMT plus-end, proximal SPB, and distal SPB were tracked throughout the 80 time points. Their kymograph traces are colored in green, magenta, and blue, respectively. **(E)** The

extracted 3D aMT length of each time point was plotted as a function of time, shown as gray dots. For this particular aMT, the maximum aMT length was 2.3 μm, and the lifetime was the full image acquisition window of 85.6 s. Two growth phases (green line over gray dots) and one shrinkage phase (magenta line over gray dots) were observed. The speeds of growth and shrinkage were calculated for each phase by dividing the corresponding Δlength by the Δtime. **(F–I)** Quantification results are shown. **(F)** All maximum lengths of microtubules below the detection limit (666.7 nm owing to the microscope resolution) were marked as 0 μm. See Materials and methods for details. The dashed (solid) bar represents the median (mean), the box marks the interquartile range, and the vertical line covers the 95% confidence interval ($n$ = 3 independent clones with a total of >400 cells per genotype analyzed). Statistical significances were calculated using one-way ANOVA, **** $P < 0.0001$, n.s., not significant. **(G)** Maximum aMT length shown as histogram, mean of length shown in red (mean ± SD of mean). **(H)** The lifetime of aMTs shown as cumulative distribution, statistical significance was assessed with the Kolmogorov-Smirnov test. **(I)** Speeds of microtubule growth and shrinkage. Statistical significances were calculated using two-tailed Student's $t$ test unless otherwise specified. **** $P < 0.0001$; * $P < 0.05$; n.s., not significant. Source data are available in Data S1.

et al., 2015). Loss of this region resulted in substantially longer astral microtubules (2.9 ± 0.2 μm, mean ± SD, $n$ = 3 independent clones, minimally 400 microtubules per clone, $P < 0.0001$) in comparison with those observed in wild-type cells (2.1 ± 0.1 μm, mean ± SD, $n$ = 3; Fig. 1, C and F). Thus, Kip2's N-terminal region was not necessary for promoting microtubule growth in vivo but rather regulated it negatively.

The C-terminal, non-motor domain (NMD) of Kip2 (Kip2-NMD, residues 504–706) contains at least one coiled-coil domain that ensures Kip2 homodimerization and is hence required for Kip2 function. Sequence analysis (Wolf et al., 1997) suggested that its last 62 residues (644–706, denoted the tail domain, T) are unlikely to form a coiled-coil structure and might not contribute to Kip2 dimerization (Fig. 1 B). Removing this domain (*kip2-ΔT* allele, Kip2 residues 1–645) caused the mutant cells to form short microtubules (Fig. 1, C and F), the dynamics of which recapitulated those of the *kip2Δ* mutant cells (Table S1). Furthermore, the *Kip2-ΔN-ΔT* mutant cells, lacking both Kip2's N-terminus and tail domains also behaved like the *kip2Δ* and *kip2-ΔT* single-mutant cells (Fig. 1, C and F; and Table S1). Thus, the C-terminal tail domain of Kip2 is essential for its microtubule polymerase activity.

Removing the motor domain (residues 100–503) caused the cells to form very short microtubules, similar to those of the *kip2Δ* mutant cells (Fig. 1, C and F). This result could simply reflect the loss of motility and thus, the low amounts of Kip2 observed on microtubule plus-ends in this mutant compared to wild-type cells (Fig. S1 C). It could also indicate a role for the motor domain in microtubule polymerization. To distinguish between these two possibilities, we tested whether restoring the levels of the protein at microtubule tips by inserting the motor domain of Kip3 in place of the endogenous one would restore microtubule growth. Thus, we generated a Kip3-Kip2 chimera protein by fusing the motor domain of Kip3 to the Kip2-NMD (residues 504–706; Fig. 1 A) and expressed it from the *KIP2* locus. This chimera accumulated abundantly at the plus-ends of astral microtubules (Fig. S1 D and Video 1), but the microtubules remained short (0.3 ± 0.1 μm, mean ± SD, $n$ = 3 independent clones, $P < 0.0001$; Fig. 1 F). We conclude that the motor domain of Kip2 (Kip2-MD) might be directly involved in microtubule polymerization.

### The C-terminal tail of Kip2 is neither required for free tubulin recruitment nor Kip2 motility
We next asked how the C-terminal tail of Kip2 contributes to microtubule growth. We reasoned that it could promote

microtubule growth by directly interacting with free tubulin dimers or microtubules, by supporting the motility of the kinesin toward microtubule tips, or through its interaction with a cofactor, such as Bik1. Thus, we expressed in bacteria and purified the Kip2-NMD (residues 504–706; Fig. 2 A and Fig. S1 E), which contains the tail of Kip2, and tested its ability to bind free tubulin and microtubules. Size-exclusion chromatography coupled to multi-angle light scattering (SEC-MALS) experiments did not reveal any detectable binding between Kip2-NMD dimers and tubulin under the conditions used (Fig. 2 B). No co-sedimentation of Kip2-NMD with Taxol-stabilized microtubules was detected either (Campbell and Slep, 2011; Fig. 2 C). Thus, the C-terminal tail of Kip2 is unlikely to directly promote either Kip2 interaction with microtubules or the recruitment of free tubulin dimers.

We next tested whether Kip2's tail influences the movement of the protein along microtubules. We drew kymographs from time-lapse series of Kip2-3xsfGFP and Kip2-ΔT-3xsfGFP speckles moving along preanaphase cytoplasmic microtubules. As reported previously (Chen et al., 2019b), Kip2-3xsfGFP speckles moved from the spindle pole bodies (SPBs) toward the plus-end of microtubules (Fig. S2 A). Unlike the wild-type protein, Kip2-ΔT-3xsfGFP speckles appeared at random places along the microtubule shaft (Fig. S2 A). Yet, their speeds did not deviate from that of the wild-type protein (6.7 ± 2.3 μm min$^{-1}$, mean ± SD, $n$ = 107 speckles vs. 6.3 ± 2.1 μm min$^{-1}$, mean ± SD, $n$ = 192 speckles in wild-type cells, $P < 0.01$, tested with unpaired $t$ test; Fig. S2 B). Thus, the C-terminal tail of Kip2 is dispensable for proper movement of the Kip2 motor along microtubules.

### The interaction between Kip2 and Bik1 requires the C-terminal tail of Kip2
Previous studies established that Bik1 and Kip2 interact directly with each other and that this interaction involves the NMD-Kip2 (Carvalho et al., 2004; Roberts et al., 2014). To confirm and extend this observation, we recombinantly expressed and purified the coiled-coil domain of Bik1 (Bik1-CC, residues 182–396; Fig. 2 A; Stangier et al., 2018). The oligomerization states and interaction capability of Bik1-CC and Kip2-NMD were assayed using SEC-MALS. In agreement with previous findings (Stangier et al., 2018), SEC-MALS analysis yielded a molecular mass for Bik1-CC of 51 kD, consistent with Bik1-CC forming a dimer in solution (calculated molecular mass of the Bik1-CC monomer: 25.5 kD; Fig. 2 D). SEC-MALS analysis of Kip2-NMD yielded a molecular mass of 45 kD, consistent with Kip2-NMD dimerizing as well (calculated molecular mass of the Kip2-NMD monomer: 23.3 kD; Fig. 2 D). When both dimers were mixed, the observed

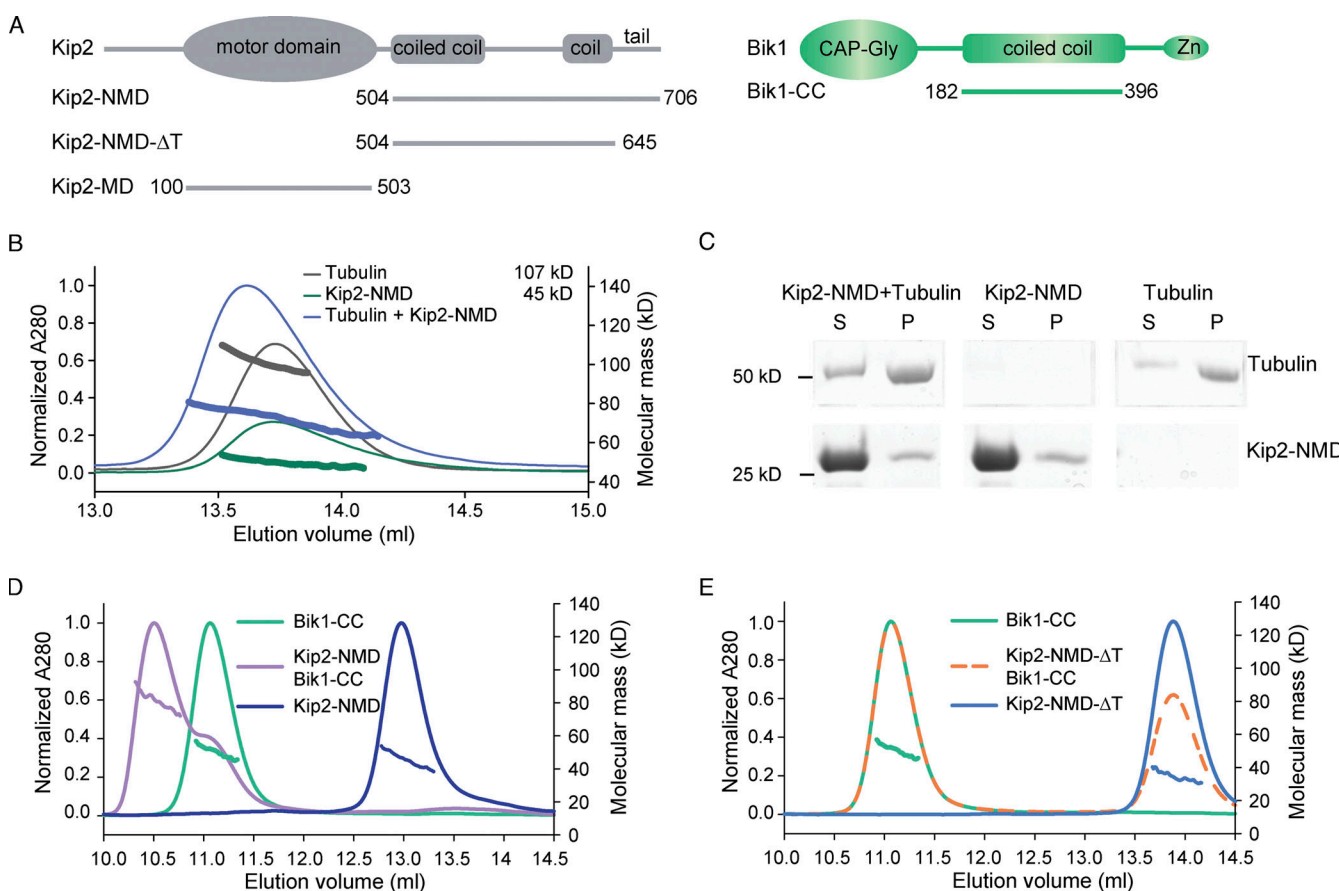

Figure 2. **Strong interaction between Kip2 and Bik1 requires the C-terminal tail of Kip2. (A)** Schematic representations of protein constructs used in this study. Zn, Zinc knuckle domain. **(B)** SEC-MALS experiments of Kip2-NMD (green; calculated molecular mass of the monomer: 23 kD), tubulin (dark gray; calculated molecular mass of αβ-tubulin: 110 kD), and a mixture of Kip2-NMD and tubulin (blue). Note that the molecular mass distribution under the Kip2-NMD/tubulin mixture elution profile is in between the ones of the Kip2-NMD dimer and the tubulin heterodimer, indicating no interaction between the two proteins. **(C)** Coomassie-Blue-stained SDS-PAGE of microtubule pelleting assays using Taxol-stabilized microtubules mixed without and with Kip2-NMD. S, supernatant; P, pellet. The supernatant lane of the Kip2-NMD alone sample is also shown in Fig. S1 E. **(D)** SEC-MALS analyses of Bik1-CC (green), Kip2-NMD (dark blue), and a mixture of Bik1-CC and Kip2-NMD (purple). **(E)** SEC-MALS analyses of Bik1-CC (green), Kip2-NMD-ΔT (light blue), and a mixture of Bik1-CC and Kip2-NMD-ΔT complex (orange). The UV absorption at 280 nm and the molecular masses across the peak determined by MALS are plotted. Source data are available for this figure: SourceData F2.

molecular mass was of 82 kD for molecular species eluting at 10.5 ml from the SEC column (Fig. 2 D). This suggests the formation of a complex composed of one Bik1-CC dimer and one Kip2-NMD dimer (calculated molecular mass of a 1:1 complex: 97.6 kD). Kip2-NMD-ΔT (lacking the C-terminal tail domain, T, of Kip2) also assembled as a dimer in vitro (measured molecular mass of 34 kD; calculated molecular mass of the Kip2-NMD-ΔT monomer: 16.5 kD; Fig. 2 E), consistent with the coiled-coil element being intact, but failed to interact with Bik1-CC as revealed by SEC-MALS (Fig. 2 E). Thus, Bik1-CC interacts with Kip2-NMD and this interaction depends on Kip2's C-terminal tail.

## Bik1 reduces the rate of Kip2 dissociation from microtubule plus-ends

Kip2-3xsfGFP decorates cytoplasmic microtubule lattices and is enriched at their plus-ends (Fig. 3 A and Chen et al., 2019b). Compared to wild-type cells, disrupting the interaction between Kip2 and Bik1 by deleting the *BIK1* gene or by removing the C-terminal tail of Kip2 (Kip2-ΔT) had several effects. First,

instead of accumulating as a nicely focused pick of intensity at the plus-end of astral microtubules, as wild-type Kip2 does in wild-type cells (Fig. 3 A), in the mutant cells the motor protein spread along the shaft of the microtubule and its intensity increasing progressively toward the plus-end (Fig. 3 A). Second, in these mutant cells, Kip2 was present overall in higher amounts along the entire shaft of astral microtubule (Fig. 3 A). In contrast, these deletions did not perturb the speed at which Kip2 speckles moved along microtubules (Fig. S2, A and B). These data suggest that Bik1 binding may control Kip2 distribution along microtubules.

To investigate this possibility, we quantified the distribution of Kip2-3xsfGFP along cytoplasmic microtubules in the presence and absence of Bik1, using Spc42-mCherry as the microtubule minus-end marker. Image data were collected for at least 500 preanaphase cells per strain. Kip2-3xsfGFP fluorescence along microtubules was extracted by line scanning analysis and aligned using the peak of the Spc42-mCherry signal at the minus-end. The average distribution of GFP fluorescence along

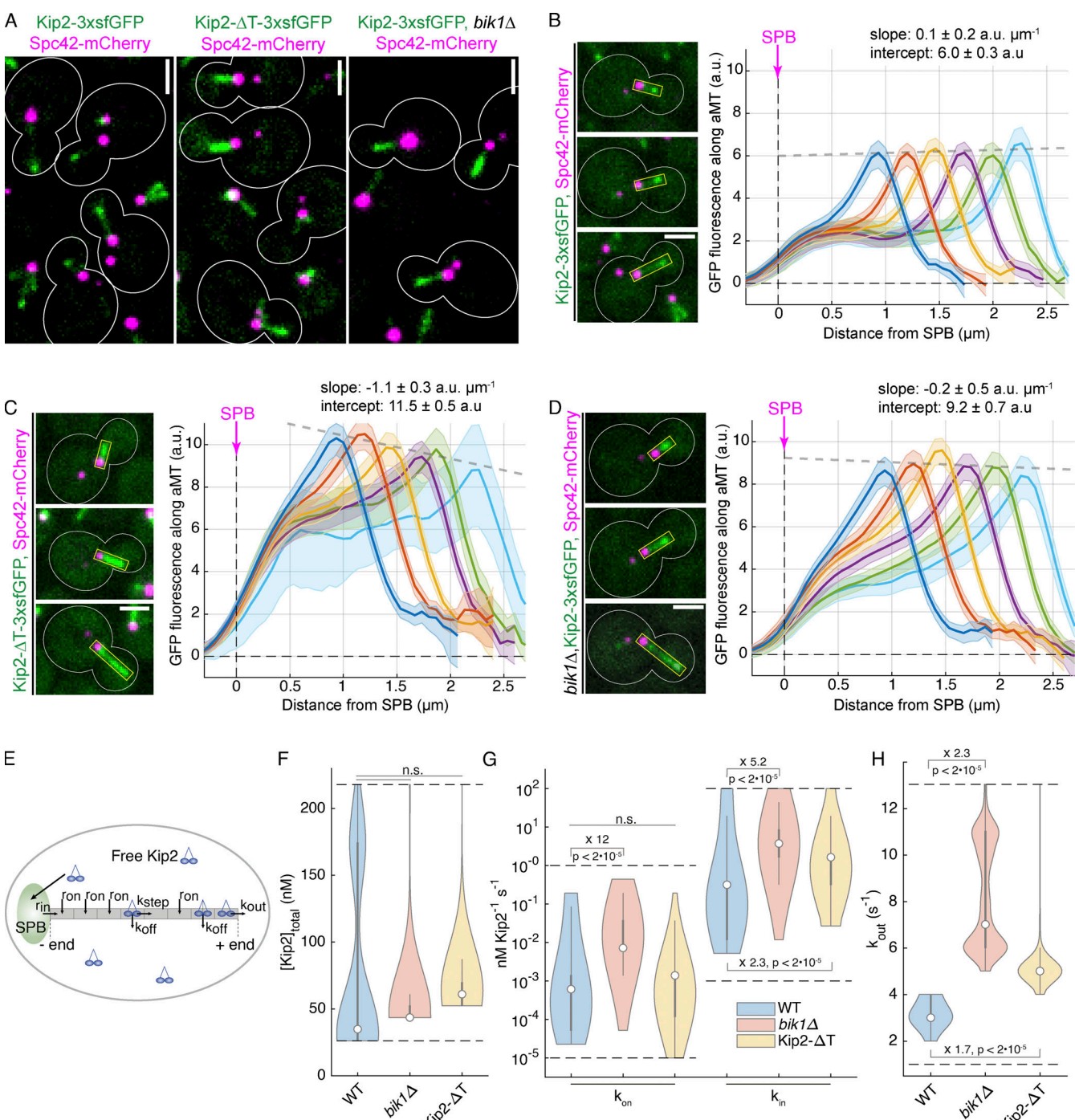

Figure 3. **Kip2–Bik1 interaction is required to retain Kip2 on cytoplasmic microtubule plus-ends efficiently. (A)** Representative images of preanaphase cells expressing Spc42-mCherry (magenta) and Kip2-3xsfGFP (green) or Kip2-ΔT-3xsfGFP (green) in the presence or absence of Bik1. **(B–D)** Representative images (left) and quantifications (right) of fluorescence intensities (a.u.) from endogenous Kip2-ΔT-3xsfGFP (C) and Kip2-3xsfGFP in the presence (B) or absence (D) of Bik1 along preanaphase cytoplasmic microtubules (boxed areas). Signals were aligned to SPBs using the Spc42-mCherry (magenta) intensity peak and binned by microtubule length (2 pixel = 266.7 nm bin size). Colored lines show mean Kip2-3xsfGFP fluorescence per bin and shaded areas represent 95% confidence intervals for the mean. Gray dashed lines denote weighted linear regressions for the mean GFP fluorescence on plus-ends over all bins. Scale bars, 2 µm. 53 ≤ $n$ ≤ 180 per bin. These graphs have consistent scales for direct comparison and per-bin comparisons are shown in Fig. S3. B has been published as Fig. 1 B in Chen et al. (2019b). **(E)** Scheme of the mathematically estimated parameters (Chen et al., 2019b). Free Kip2 (with concentration $[Kip2]_{free} < [Kip2]_{total}$) binds to the microtubule minus-end anchored at the SPB with rate $r_{in} = k_{in}[Kip2]_{free}$ if the minus-end site is free and to any free lattice site with rate $r_{on} = k_{on}[Kip2]_{free}$. A bound motor can detach with rate $k_{off}$, and it can advance with rate if the following site toward the plus-end is free. At the plus-end, the motor detaches with a different rate, $k_{out}$. **(F)** Likelihood of total Kip2 concentration $[Kip2]_{total}$ estimated from fit in B–D. **(G)** Likelihood of on rate constant $k_{on}$ and in rate constant $k_{in}$ estimated from fit in B–D. Statistical significance for differences determined by sampling from the likelihood (see supporting information), the difference in the median as indicated. **(H)** Likelihood of out rate $k_{out}$ constant estimated from fit in B–D. For F–H, median values are indicated as circles, interquartile range (IQR) by thick gray bars, and 1.5×IQR by thin gray bars. Kernel density estimates are computed from 20,000 samples from the likelihood function. Dashed black lines indicate sampled parameter ranges.

cytoplasmic microtubules was then computed as a function of microtubule length as reported previously (Chen et al., 2019b; Fig. 3, B–D). In contrast to wild-type cells (Fig. 3 B; Chen et al., 2019b), where it peaks sharply at the plus-end of microtubules, the Kip2-3xsfGFP signal in bik1Δ cells increased along the entire microtubule without forming much of a distinct peak at microtubule ends. This distribution suggests that in these cells, Kip2 is also recruited along microtubule shafts and not only at the minus-end of microtubules and does not make a pause at the plus-ends of microtubules, two key features of its behavior in wild-type cells (Chen et al., 2019b). A similar pattern was observed in cells expressing Kip2-ΔT-3xsfGFP.

To characterize in more detail how the removal of Bik1 affected the behavior of Kip2, we next applied a previously established mathematical model for estimating how Kip2 moves along microtubules (Chen et al., 2019b), based on the imaging data and the physical law of conservation of matter. According to this assumption, the Kip2 dynamics can be described through only five parameters: its recruitment at microtubule minus-ends ($k_{in}$, rate constant for landing and starting a run from the minus-end) and lattices ($k_{on}$, rate constant for landing and starting a run at a location on the microtubule lattice), Kip2 dissociation from microtubule lattices (rate constant $k_{off}$), the motor's stepping speed (rate constant $k_{step}$), and Kip2 detachment from microtubule plus-ends (rate constant $k_{out}$; Fig. 3 E; Chen et al., 2019b). We used the in vivo experimental distributions of Kip2-3xsfGFP to infer how abrogating Kip2 interaction with Bik1 may affect microtubule dynamics (Fig. 3, F–H, see supporting information and Chen et al., 2019b). Importantly, we use this method in order to estimate parameter values and not primarily to generate predictions. With a Bayesian approach, we also estimate the uncertainties of parameter values; these uncertainties may be caused by measurement noise or by limited identifiability of individual parameters. Good matches between experimental data and simulation results (Fig. S3, G–I, and below), except for peak densities in Kip2-ΔT-3xsfGFP cells, which are slightly off, indicate that the approach is suitable for estimating parameters from our in vivo data.

In vitro, Kip2 resides at microtubule ends for ~30 s and accumulates there (Hibbel et al., 2015). Since this is the site at which microtubule growth takes place, we first wondered whether removing Bik1 affected the time Kip2 resides at microtubule tips in vivo. Our $k_{out}$ estimates imply that in wild-type cells, Kip2 remains bound to microtubule tips for much shorter periods of time than in vitro, <0.3 s. Interestingly, simulations with in vitro $k_{out}$ values (Hibbel et al., 2015) showed clear inconsistencies with our experimental data, demonstrating that Kip2 lingers for shorter times at the end of microtubules in vivo than in vitro. In particular, using the $k_{out}$ values measured in vitro, we predict "traffic jams" at microtubule plus-ends (Fig. S3 J), which we do not observe in vivo. The plus-end residence times of the related kinesin-8 Kip3 decrease exponentially with (total) Kip3 concentration (Varga et al., 2009) and with applied mechanical force (Bugiel et al., 2020). We hypothesize that Kip2 behaves similarly. This would explain the different residence times observed in vitro (with <1 nM total Kip2; Hibbel et al., 2015) and in vivo (40 nM and above; Fig. 3 F). It would also be

consistent with decreasing peak densities (slopes in Fig. 3, B–D) for (long) microtubules with high Kip2 density (intercepts in Fig. 3, B–D), and with model mismatches particularly for long astral microtubules (aMTs) in Kip2-ΔT-3xsfGFP cells (Fig. S3 I) where such concentration effects are most pronounced (Kip2-ΔT is ~50% more abundant than the wild-type protein, although removing Bik1 has little effect on Kip2 levels; Fig. 3 F and Fig. S2 D). The low residence time of Kip2 at microtubule plus-ends in vivo might reflect competition with other +TIPs and motor proteins. Irrespective of these considerations, our estimates indicated that the out-rate constant of Kip2 at the microtubule plus-end is significantly increased in bik1Δ mutant cells and in Kip2-ΔT-3xsfGFP cells compared to the wild-type protein in wild-type cells (Fig. 3 H). Thus, when they cannot interact with Bik1, individual Kip2 molecules leave the microtubule plus-ends essentially as they reach it.

In addition to this, our estimates for Kip2's on-rate constants were orders of magnitude lower than its in-rate constants, as previously observed (Chen et al., 2019b). Thus, in wild-type cells and both mutants, Kip2 is much more likely to start a run along the microtubule at its minus end than at any other position on the microtubule. Correspondingly, disrupting the Bik1–Kip2 interaction did not dramatically affect Kip2 recruitment at the microtubule minus-end, i.e., the SPB. The estimates, however, indicated as well that Kip2's on-rate and in-rate constants were both significantly increased in bik1Δ and to a lesser extent in the Kip2-ΔT-3xsfGFP mutant cells (Fig. 3 G), consistent with the accumulation of Kip2-3xsfGFP along cytoplasmic microtubules in these cells (Fig. 3, B and D). This is also consistent with the observation that both Kip2-ΔT-3xsfGFP in BIK1 wild-type cells and Kip2-3xsfGFP in bik1Δ mutant cells assemble speckles at random places along microtubules, unlike Kip2-3xsfGFP in wild-type cells (Fig. S2 A). We concluded that Bik1 facilitates the retention of Kip2 at microtubule plus-ends, while inhibiting its recruitment onto microtubules at SPBs and along microtubule shafts.

**The Kip2-NMD binds to SPBs and tracks microtubule plus-ends by binding to Bik1**

To investigate further the notion that Bik1 retains Kip2 at microtubule plus-ends, we characterized the localization of Kip2-NMD-3xsfGFP in vivo. Kip2-NMD-3xsfGFP binds Bik1 and lacks Kip2-MD. Strikingly, while the majority of Kip2-NMD-3xsfGFP expressed from the endogenous KIP2 locus localized diffusely throughout the cytoplasm, enrichment of the GFP signal was also observed in few foci (Fig. 4, A–C). Co-expression of fluorescent SPB markers, Spc42-mCherry (Fig. 4 A) or Spc72-GFP (Fig. 4 C), indicated that these foci corresponded to SPBs and the plus-end of cytoplasmic microtubules. Furthermore, time-lapse images show that Kip2-NMD-mNeonGreen tracks the plus-end of both growing and shrinking microtubules (Fig. 4 C and Video 2). Loss of Bik1 abolished Kip2-NMD-3xsfGFP localization to SPBs and microtubule plus-ends (Fig. 4, A and B). Furthermore, Kip2-NMD-ΔT-3xsfGFP, which cannot interact with Bik1, failed to accumulate at SPBs and microtubule plus-ends (Fig. 4, A and B). Loss of the kinesin Kip3, which like Kip2 localizes to microtubule plus-ends, had no such effect. We concluded that Bik1

Figure 4. **Kip2-NMD binds to SPBs and tracks microtubule plus-ends by binding to Bik1. (A)** Representative images of preanaphase cells expressing Spc42-mCherry (magenta) and Kip2-NMD-3xsfGFP (green) or Kip2-NMD-ΔT-3xsfGFP (green) in the context of control, *bik1Δ*, or *kip3Δ*. **(B)** Quantification of Kip2-NMD-3xsfGFP or Kip2-NMD-ΔT-3xsfGFP localization at preanaphase SPBs and aMT plus-ends in control and mutant cells. Cells from three independent clones were quantified, *n* as shown in the graph. **(C)** Kip2-NMD-mNeonGreen accumulates on both the growing and shrinking cytoplasmic microtubule plus-ends. The red arrowhead marks the plus-end. SPBs are visualized with Spc72-GFP. The time-lapse movie has 1.07 s intervals, shown as Video 2; only representative time points are shown here. **(D)** Representative images of preanaphase heterozygous diploid cells expressing Kip2-G374A-3xsfGFP (green) and the wild-type protein Kip2-mCherry (magenta). Line scan analysis of the two aMTs as highlighted in yellow boxes on the right. All scale bars, 2 μm.

binds Kip2-NMD at SPBs and microtubule plus-ends. Together, these data indicate that Bik1 facilitates the retention of Kip2 at the location where it is itself enriched.

To investigate whether Bik1 also recruits soluble full-length Kip2 to SPBs and microtubule plus-ends, we tested whether the motor activity of the full-length Kip2 protein is required for its retention at these locations by imaging both haploid cells expressing the ATPase-deficient protein fused to 3xsfGFP (Kip2-G374A-3xsfGFP) and heterozygous diploid cells expressing both Kip2-G374A-3xsfGFP and the wild-type protein fused to mCherry (Kip2-mCherry; Fig. 4 D and Fig. S2, E and F). Consistent with previous results (Chen et al., 2019b), the Kip2-mCherry protein decorated microtubule shafts, SPBs, and plus-ends. In contrast, Kip2-G374A-3xsfGFP was recruited to SPBs, but not microtubule shafts and plus-ends, whether Kip2-mCherry was

present or not. At SPBs, the levels of Kip2-G374A-3xsfGFP were reduced by 30% upon deletion of the *BIK1* gene (Fig. S2, E and F). These results establish that Bik1 somewhat facilitates the recruitment of cytoplasmic, full-length Kip2 to SPBs but cannot do so at microtubule plus-ends. Instead, full-length Kip2 requires its motor activity along microtubule shafts to reach microtubule plus-ends, where Bik1 increases its retention time.

### The Kip2-MD binds free tubulin dimers
The data collected so far suggest that Kip2 promotes microtubule growth when retained at the plus-end of the microtubule. Furthermore, our data suggest that the motor domain is essential for microtubule growth. Therefore, we wondered whether the motor domain could act directly in microtubule polymerization. To consider this possibility further, we tested whether the motor domain

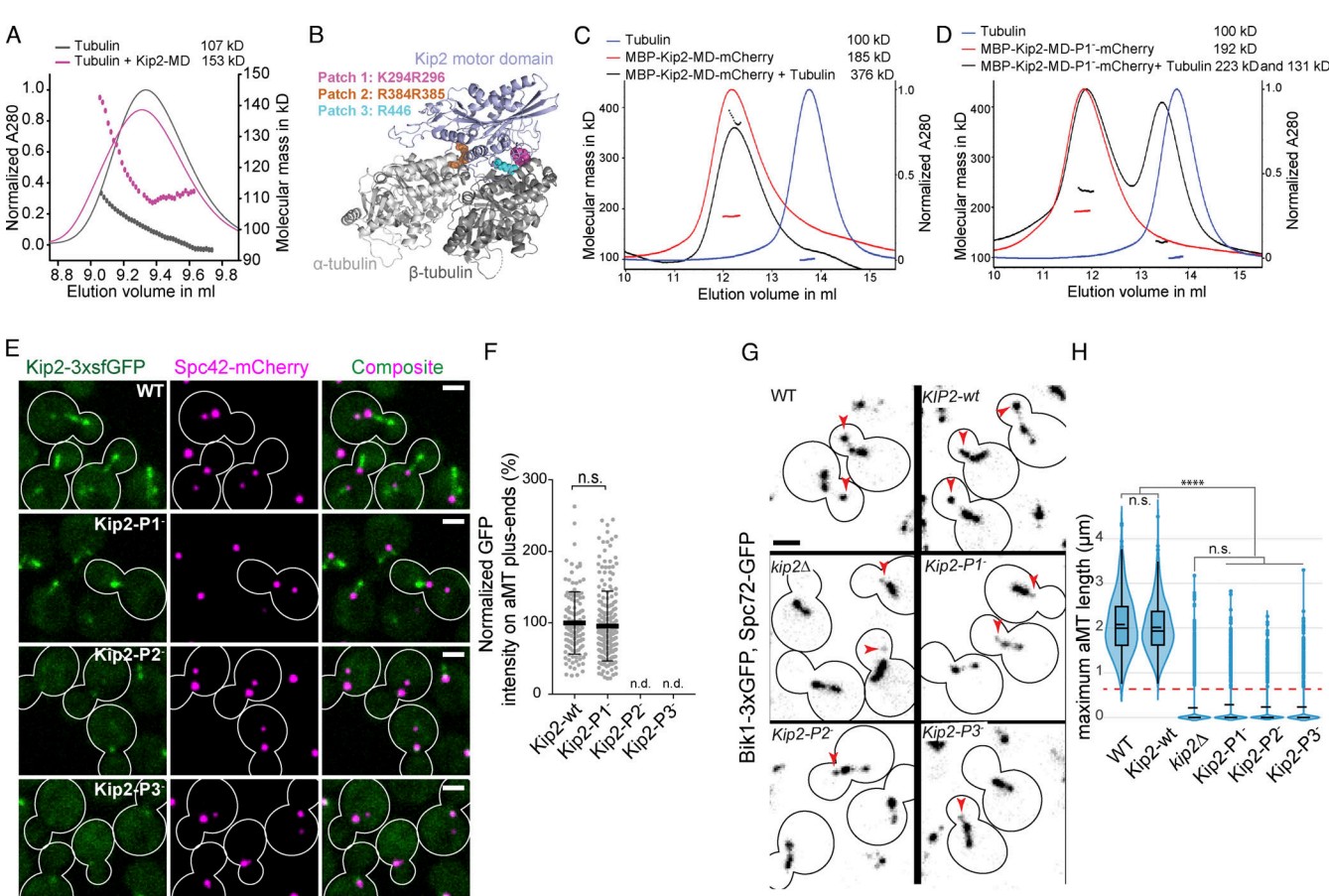

Figure 5. **The motor–tubulin interactions underlie the microtubule polymerase activity of Kip2. (A)** SEC-MALS analysis of a mixture of apo Kip2-MD (calculated molecular mass of the monomer: 44 kD) and tubulin (calculated molecular mass of the dimer: 110 kD). **(B)** Homology model of the Kip2-MD (blue) in complex with the αβ-tubulin heterodimer (gray). The three positively charged Kip2-MD surface residue patches crucial for tubulin or microtubule-binding are highlighted in different colors (see corresponding legend). **(C)** SEC-MALS analysis of a mixture of MBP-Kip2-MD-mCherry (calculated molecular mass of the monomer: 125.8 kD) and tubulin. **(D)** SEC-MALS analysis of a mixture of MBP-Kip2-MD-P1⁻-mCherry (same calculated molecular mass as MBP-Kip2-MD-mCherry) and tubulin. **(E)** Representative images of metaphase yeast cells expressing Spc42-mCherry (magenta) and wild-type Kip2 or Kip2 containing mutations in tubulin-binding patches C-terminally fused with 3xsfGFP (green). **(F)** Quantification of GFP fluorescence intensities at aMT plus-ends from cells shown in E, error bars represent mean ± SD (*n* = 3 independent clones, with a total of >100 cells per genotype analyzed). For P2 and P3 mutant cells, the GFP signal was only weakly associated with SPBs; therefore, in both cases, the intensities on aMT plus-ends were not determined (n.d.). The statistical significance (n.s.: not significant) was test with two-tailed Student's *t* test. **(G and H)** Quantification of 3D length of aMTs. Kip2-wt represents the control of inserting the selection marker TRP1 downstream of the KIP2 gene; see the Materials and methods section for more details. Representative images of metaphase cells of indicated genotype are shown in G; red arrowheads mark the plus-ends of aMTs. Quantification results are shown in H, the dashed (solid) bar represents the median (mean), the box marks the interquartile range, and the vertical line covers a 95% confidence interval (*n* = 3 independent clones, with a total of >400 cells per genotype analyzed). Statistical significances were calculated using one-way ANOVA; **** P < 0.0001; n.s., not significant. Scale bars, 2 μm. Source data for F and H are available in Data S1.

had a detectable affinity for free tubulin dimers. To this end, we expressed and purified the Kip2-MD (residues 100–503; Fig. 2 A) from bacteria and carried out SEC-MALS analysis of an equimolar mixture of tubulin and Kip2-MD in its apo state. As shown in Fig. 5 A, these two proteins assembled in a complex of a molecular mass of 153 kD, indicating the formation of a 1:1 complex between tubulin and Kip2-MD (calculated molecular masses for αβ-tubulin and Kip2-MD: 110 and 44 kD, respectively). Thus, Kip2-MD exhibits detectable affinity for free tubulin dimers in vitro.

**The Kip2-MD-tubulin interaction underlies the microtubule polymerase activity of Kip2 in vivo**

To investigate the structural basis of the interaction between Kip2 and free tubulin, we developed a homology model of its

motor domain bound to a tubulin dimer in its "curved" conformation, based on a published high resolution complex structure formed between the motor domain of Kif2C and tubulin (Wang et al., 2017). The homology model predicted three positively charged surface residue patches to be potential tubulin-binding sites. The three patches corresponded to residues Lys294 and Arg296 (patch 1, P1), Arg384 and Arg385 (patch 2, P2), and Arg446 (patch 3, P3; Fig. 5 B and Fig. S4 A). Classical motile kinesins expose two tubulin binding sites, one that makes contact with α-tubulin and the other with β-tubulin (Gigant et al., 2013). Interestingly, in our model, both P1 and P3 are predicted to bind β-tubulin (Fig. 5 B and Fig. S4 A). Thus, we wondered whether these patches could discriminate between the two possible conformations of tubulin, namely, the straight

conformation found in the microtubule shaft and the curved conformation of the free dimer. To investigate the roles of the three patches for Kip2 function, we first substituted the positively charged residues of each patch with alanines in yeast cells.

As described previously (Caudron et al., 2008; Carvalho et al., 2004), Kip2-3xsfGFP decorates astral microtubules and accumulates at their plus-ends (Fig. 5 E). The localization of Kip2-3xsfGFP also held in heterozygous diploid cells, in which the other copy of Kip2 was fused to mCherry (Kip2-mCherry; Fig. S4 C). Interestingly, the Kip2-3xsfGFP bearing mutations in P2 or P3 (denoted Kip2-P2⁻-3xsfGFP and Kip2-P3⁻-3xsfGFP) localized diffusively throughout the cytoplasm and barely decorated astral microtubules. These mutant proteins only associated, albeit weakly, with SPBs in haploids (Fig. 5, E and F) and faintly with astral microtubules in heterozygous diploids, possibly by dimerizing with Kip2-mCherry (Fig. S4 C). In contrast, the P1 mutant Kip2-P1⁻-3xsfGFP localized properly to astral microtubules and accumulated at their plus-ends to the same level as the wild-type protein in both haploid and diploid cells (Fig. 5, E and F; and Fig. S4 C). We concluded that the residues of both P2 and P3 are essential for Kip2 binding to microtubule lattices and its translocation along them. In contrast, the residues of P1 were dispensable for these processes.

Strikingly, the astral microtubules decorated with Kip2-P1⁻-3xsfGFP in the preanaphase cells appeared overall much shorter than those of wild-type cells (Fig. 5, E, G and H), suggesting that despite its proper localization, the Kip2-P1⁻-3xsfGFP protein was unable to promote microtubule growth. To systematically evaluate the effect of this patch mutation on microtubule polymerization in haploid cells, we measured its effects on the 3D length of the astral microtubules (as in Fig. 1, C and D). In this assay, all three patch mutants phenocopied the effects of kip2Δ mutation (Fig. 5, G and H). The remarkably short astral microtubules of the P1⁻ mutant cells (0.3 ± 0.01 µm, mean ± SD, P < 0.0001) confirmed that the Kip2-P1⁻ molecules, which accumulated at their plus-ends, are unable to promote the polymerization of the astral microtubules. These results indicate that Kip2 relies on a putative tubulin-binding patch located directly in its motor domain in order to promote microtubule polymerization.

To confirm that the P1 mutations indeed impair tubulin binding by Kip2, we performed in vitro experiments with its motor domain. In contrast to the wild-type Kip2-MD (see above), we were not able to produce a well behaved P1 mutant. However, including a first helix predicted to form part of the coiled coil and fusing MBP to the N-terminus and mCherry to the C-terminus of the motor domain resulted in soluble and stable proteins (denoted MBP-Kip2-MD-mCherry and MBP-Kip2-MD-P1⁻-mCherry, see Table S3 for details of the constructs). To study the interactions of the two proteins with tubulin, we performed SEC-MALS experiments. These analyses revealed molecular masses for MBP-Kip2-MD-mCherry and MBP-Kip2-MD-P1⁻-mCherry of 185 and 192 kD, respectively (Fig. 5, C and D; and Fig. S1 E), consistent with the presence of mixed monomers and dimers (calculated molecular mass of each monomer: 125.8 kD). It also revealed the formation of a complex composed of one MBP-Kip2-MD-mCherry dimer and one tubulin dimer (calculated molecular mass of a tubulin dimer: 110 kD) with a

molecular mass of 376 kD (calculated molecular mass of each tetramer: 361.6 kD). For MBP-Kip2-MD-P1⁻-mCherry, we obtained elution profiles and molecular masses suggesting a significantly weaker interaction of the Kip2 mutant with tubulin (Fig. 5 D). Thus, the Kip2 motor domain indeed binds unpolymerized tubulin and P1 facilitates this interaction.

### The Kip2 patch 1 is essential for Kip2's microtubule polymerization activity in vitro

To test how direct the role of P1 in promoting microtubule growth is, the polymerization activities of wild-type Kip2 and Kip2-P1⁻ were measured in dynamic microtubule assays using porcine tubulin in the presence of ATP (Hibbel et al., 2015). At 2 nM concentration, purified wild-type MBP-Kip2(1-560) fused to RFP (Fig. 6, A and C; Table S2, and Fig. S1 E, denoted Kip2-WT) strongly increased the length of freshly polymerized microtubules by nearly fivefold (P < 0.0001, t test). At the same buffer and tubulin concentrations, MBP-GFP alone rarely caused microtubule growth. Under the same conditions, MBP-Kip2(1-560)-P1⁻-RFP (denoted Kip2-P1⁻) did not demonstrate any microtubule polymerizing activity either. Doubling or quadrupling the concentrations of Kip2-P1⁻ slightly increased microtubule growth but still four- to fivefold less than with Kip2-WT (P < 0.0001; Fig. 6, A and C; and Table S2). Quantification of the kymographs drawn from the time-lapse images (Fig. 6 B) indicate that a 2 nM concentration of Kip2-WT increased the average speed of microtubule growth (the slope of the growing microtubule in the kymograph) at least 13-fold and reduced the frequency of catastrophe about 19-fold (Fig. 6, D and E; and Table S2; P < 0.0001 compared to without addition). Under the same conditions, Kip2-P1⁻ failed to speed up microtubule growth and had a 10-fold milder effect on catastrophe frequency than the wild-type protein (Fig. 6, D and E; and Table S2). Doubling or quadrupling the concentrations of Kip2-P1⁻ did increase the average growth speed of microtubules up to nearly 2.9-fold, but this was still four times slower than what was obtained with the wild-type protein. Increasing Kip2-P1⁻ concentration did not change the catastrophe frequency (Fig. 6 D and Table S2). Thus, abrogation of P1 renders Kip2 nearly unable to polymerize microtubules at physiologically relevant concentrations (Chen et al., 2019b). We conclude that tubulin-binding P1 is required to polymerize microtubules both in vitro and in vivo, potentially through mediating interaction of the motor domain with free tubulin dimers.

### The Kip2 P1 is dispensable for Kip2 motility in vivo and in vitro

Based on the results above, we reasoned that P2—which mediates contact with α-tubulin (Fig. 5 B)—might be paired either with P1 or P3 to bind free tubulin dimers or microtubules, respectively. We inspected whether the P1⁻ mutations interfered with the motility of the protein along microtubules to test whether P1 mediates Kip2 interaction with microtubule lattices in vivo. Kip2 fused C-terminally with fluorescent proteins move along astral microtubule lattices as fluorescent speckles (Carvalho et al., 2004). The very short astral microtubules in Kip2-P1⁻-3xsfGFP–expressing cells made it impossible to observe any moving speckles within such short ranges. To circumvent this difficulty, we carried out this study in kip3Δ mutant cells, which form long

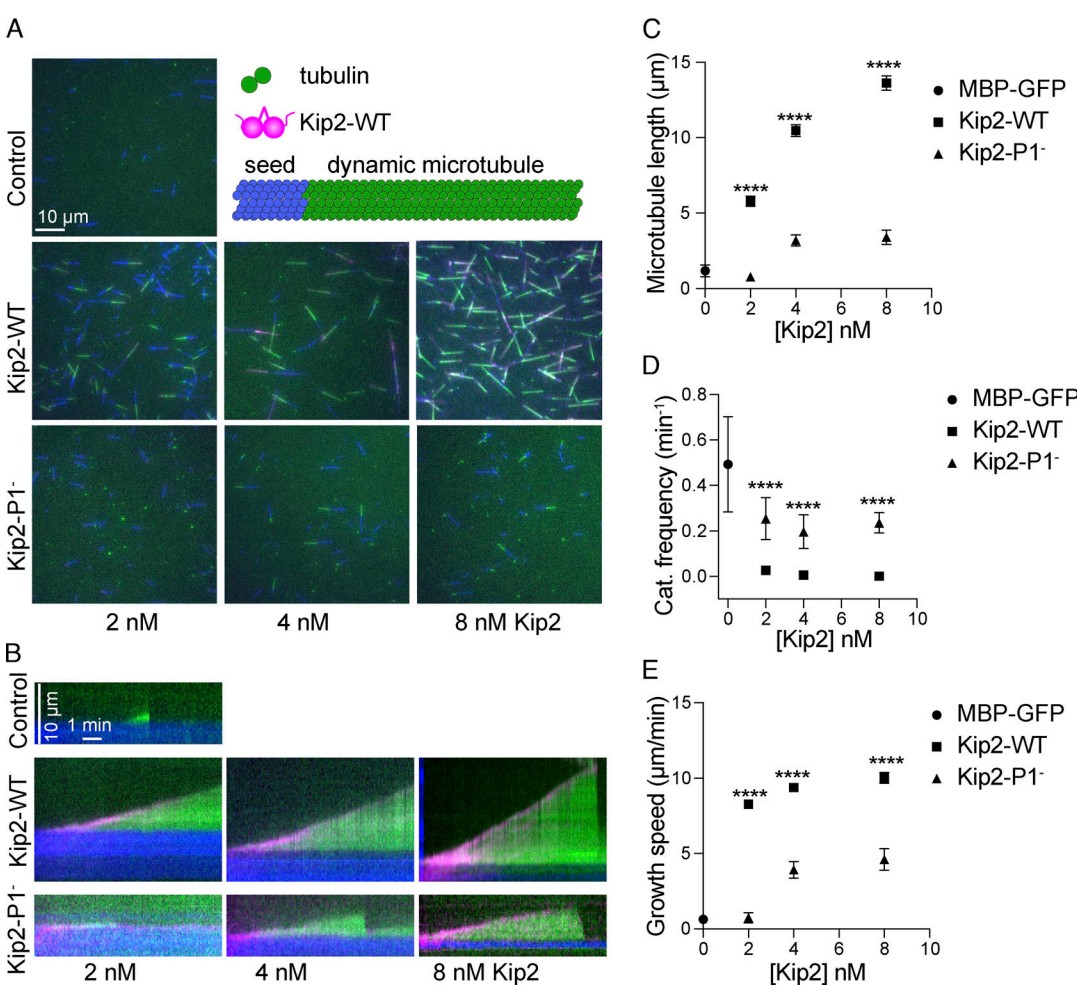

Figure 6. **Residues K294 and R296 of Kip2 are involved in interaction with free tubulin and are indispensable for efficient microtubule polymerization in vitro. (A)** Schematic of the experimental design: porcine tubulin (green) polymerizes onto stabilized microtubule seed (blue) in the presence of Kip2-WT, composed of MBP-Kip2(1-560)-RFP (magenta). And representative images of dynamic microtubules grown from seeds without (top panel) and with increasing amounts (2, 4, 8 nM) of Kip2-WT (middle panel) and Kip2-P1⁻ (bottom panel). **(B)** Kymographs drawn from time-lapse corresponding to A show microtubule growth without (top) and with increasing amounts (2, 4, 6 nM) of Kip2-WT (middle panel) and Kip2-P1⁻ (bottom). **(C–E)** Microtubule average length (C), catastrophe frequency (D), and growth speed (E) as a function of Kip2-WT or Kip2-P1⁻ concentration. These results were quantified from more than 100 growth segments per condition. Statistics significance comparing Kip2-WT and Kip2-P1⁻ were calculated with one-way ANOVA. **** P < 0.0001.

microtubules (Gupta et al., 2006; Varga et al., 2006), even if carrying the *kip2-P1⁻* allele (Fig. S5 A). This enabled us to observe plenty of Kip2-3xsfGFP or Kip2-P1⁻-3xsfGFP speckles moving toward the plus-end of astral microtubules (Fig. S5 A) and to measure their speed. Interestingly, in these cells, Kip2-P1⁻ moved slightly faster than the wild-type protein (7.1 ± 2.4 µm min⁻¹, mean ± SD, *n* = 159 speckles versus 6.5 ± 1.6 µm min⁻¹, mean ± SD, *n* = 92 speckles in control cells, P < 0.05, 25°C; Fig. S5 B). Single-molecule motility assays in vitro led to the same conclusion (Fig. S5, C and D). Kymographs of individual GMPCPP-stabilized microtubules revealed that single Kip2-WT and Kip2-P1⁻ molecules associated with the lattice at random places and walked processively toward the plus-end of the microtubules (Fig. S5 C). The average speed of Kip2-P1⁻ was 5.3 ± 2.3 µm min⁻¹ (mean ± SD, *n* = 54), slightly higher than that of Kip2-WT 4.4 ± 1.4 µm min⁻¹ (mean ± SD, *n* = 80, P < 0.01; Fig. S5, C and D). We concluded that the residues K294 and R296 are not essential for Kip2 motility in vivo and in vitro. Collectively our results reveal that the motor

domain of Kip2 has acquired a specialized surface that is dedicated to tubulin binding and polymerization.

## Discussion

Microtubule polymerases accelerate microtubule growth by increasing the local concentration of free tubulin dimers at the microtubule plus-end or the rate of tubulin incorporation (Hibbel et al., 2015). Previous studies showed that the microtubule-stabilizing kinesin Kip2 moves toward and accumulates at microtubule plus-ends in vivo (Chen et al., 2019b; Carvalho et al., 2004). In vitro, Kip2 is also a processive plus-end directed motor protein and autonomously promotes microtubule growth and inhibits catastrophe events (Hibbel et al., 2015). However, how Kip2 promotes microtubule growth was unknown. In vivo, the polymerase activity of Kip2 requires Bik1 function (Carvalho et al., 2004). Here, we identify two elements of Kip2 as essential for its microtubule polymerase activity. One is a tubulin-

interaction interface in the motor domain of Kip2. This interface is not relevant for Kip2 interaction with and its motility along microtubules but plays a key role in microtubule polymerization. The second is a binding site for the cytoplasmic linker protein Bik1 that, according to our estimates, seems to increase Kip2 residence time at microtubule plus-ends in living cells, in a Bik1-dependent manner. Based on these observations, we propose a model for how Kip2 may contribute to microtubule polymerization in vivo (Fig. 7). Whereas Kip2 might simply stabilize newly incorporated tubulin at the microtubule end, we suggest that once arrived at the very end of a microtubule, through its motile translocation along the shaft, the free motor domain of Kip2 actually binds free tubulin and promotes its incorporation in the protofilament. Thus, Kip2 might elongate its own track under its own "feet" once arrived at microtubule tips. We suggest that free tubulin binding is mediated by the positively charged patch P1 of the motor domain, whereas P2 and P3 mediate the recognition of the microtubule shaft and movement along it. This is in line with the observation that P1 is at an optimal position for reaching the helix βH12 of β-tubulin in the curved but not in the straight tubulin conformation (Fig. S4 B), suggesting that it mediates interaction with free tubulin (curved), specifically, and not with polymerized tubulin (straight; Brouhard and Rice, 2014). Therefore, although more detailed mechanistic studies will be needed to address this possibility, we speculate that the delivery and release of tubulin dimers at the elongating microtubule end is ruled by the same mechanisms as those mediating binding and release of tubulin on the shaft during motility, i.e., are controlled by the ATPase cycle of Kip2-MD. Consistent with this idea, ATP hydrolysis is essential for Kip2 to promote microtubule growth both in vitro and in vivo (Hibbel et al., 2015; Chen et al., 2019b).

A priori, several scenarios may account for how Kip2 promotes microtubule growth. First, Kip2 might transport and deliver tubulin to the microtubule plus-end, thereby increasing the local concentration of free tubulin (Hibbel et al., 2015). However, the motor domain of Kip2 cannot be involved in this, as its free-tubulin interaction interface P1 is not accessible when Kip2 is bound to the microtubule lattices. Instead, P1 would become available for tubulin binding only when at least one of the dimers' motor domains has no shaft available for binding, such as when it has passed the very end of the microtubule. Therefore, if Kip2 were to transport tubulin dimers to microtubule tips, this would have to depend on additional yet unidentified binding sites for free tubulin dimers on Kip2. Our data indicate that neither the C-terminal cargo-binding nor motor domains of Kip2 are likely to be involved in such a function. Thus, we propose that the function of these domains in tubulin polymerization takes place at the microtubule plus-end specifically and not in tubulin transport.

Second, Kip2 might also function through the transport and delivery of Bik1 to microtubule plus-ends where Bik1 might stabilize growing microtubules (Carvalho et al., 2004; Caudron et al., 2008). Supporting this idea, we confirm that Kip2 indeed supports the accumulation of Bik1 at microtubule plus-ends (Fig. S2 C). However, Kip2 can promote microtubule growth in the absence of Bik1 in vitro (Hibbel et al., 2015). Thus, Bik1 recruitment to microtubule tips is unlikely to be the ultimate mechanism by which Kip2 promotes microtubule growth.

Third, Kip2 might directly promote tubulin incorporation at microtubule plus-ends. Several lines of evidence support this last possibility. One such line concerns the role of Bik1. Indeed, our study reveals that Bik1 accumulation at microtubule plus-ends and its activity in retaining Kip2 there require Bik1's direct interaction with the Kip2-NMD and is essential for Kip2 to promote microtubule growth. Both the loss of this interaction or of Bik1 altogether decreases the time Kip2 spends at microtubule plus-ends: our out-rate constant estimates for Kip2 dissociation are consistent with Kip2 dissociating as soon as it arrives at the microtubule plus-end when Bik1 is absent. Importantly, a high flux of Kip2 arrival at the microtubule plus-end, as observed in the cells of genotype *Kip2-3xsfGFP bik1Δ* and *Kip2-ΔT-3xsfGFP BIK1* (Fig. S2 A), was not sufficient for promoting microtubule elongation. Thus, for Kip2 to promote microtubule growth, it needs to linger around the microtubule plus-end for some time, in a Bik1-dependent manner. This result is consistent with the idea that the polymerase activity of Kip2 does not involve cargo delivery but some specific activity of Kip2 at microtubule plus-ends. Together, these observations fit best with the third model, whereby Kip2 acts as a polymerase specifically at microtubule plus-ends. An interesting question, however, is why retention of Kip2 at microtubule plus-ends by Bik1 is required for microtubule polymerization in vivo but not in vitro? Possibly, the high number of plus-end binding proteins present might cause intense competition for binding at the microtubule tip in vivo and impose a requirement for an additional retention factor there. In addition, this active retention by Bik1 may alleviate the negative consequences of expressing Kip2 at high levels in vivo (Varga et al., 2009; Bugiel et al., 2020). Decreased competition in vitro could then relax this requirement. Alternatively, the availability of free tubulin might be lower in vivo than in vitro, requiring Kip2 to stay longer on microtubule tips in order to fulfill its polymerase function.

Our study also begs for a revised model of Bik1 functions in microtubule assembly. Bik1 moves along microtubule shafts with Kip2 and also appears in the vicinity of SPBs. The latter pool includes Bik1 bound to kinetochores (He et al., 2001; Lin et al., 2001), to SPB-enriched proteins including Stu2, the yeast ortholog of XMAP215/hTOG/Dis1 (Chen et al., 1998; Lin et al., 2001; Stangier et al., 2018), and to the plus-ends of very short microtubules. Although it is difficult to resolve these different sub-pools using conventional light microscopy in yeast, the reduced levels of Kip2-G374A-3xsfGFP at SPBs in the *bik1Δ* cells as well as the estimated in-rate constants suggest a previously unknown function of Bik1 also in promoting the recruitment or retention of Kip2 at SPBs. According to our parameter estimates, disruption of the Bik1–Kip2 interaction resulted in an increased entry rate of Kip2 at random sites along the shafts and from the minus-ends of microtubules and a decreased residence time of Kip2 at microtubule plus-ends. This indicates that the interaction between Bik1 and Kip2 has at least three functions. First, Bik1 promotes Kip2 recruitment to SPBs. Second, Bik1 inhibits Kip2 from starting runs from random locations along microtubule shafts. Third, Bik1 prolongs the residence time of Kip2 at microtubule plus-ends and thereby promotes the polymerase activity of Kip2 there. Of note, while Kip2-NMD binds Bik1 at

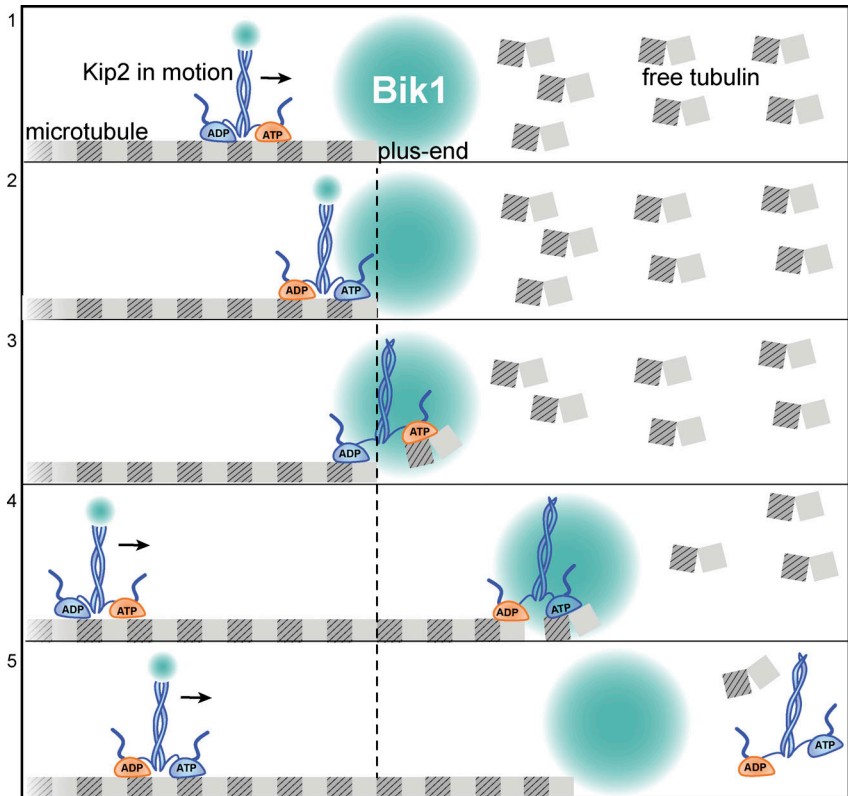

The large turquoise sphere represents the pool of Bik1 enriched at the microtubule tip.

Figure 7.   **Model of Kip2 polymerizing microtubules in living cells.** Frame 1: The two motor domains of kinesin Kip2 coordinate their ATPase cycles to move toward the microtubule plus-end. Each motor domain (blue or orange to differentiate the two) is bound to a tubulin heterodimer (gray box, β subunit; striped box, α subunit) in its straight conformation along a microtubule protofilament (the cylindrical microtubule is composed of 13 protofilaments). Bik1 is accumulated at the plus-end and is transported there by Kip2. Frame 2: Upon arriving at the plus-end, Kip2 is retained there by binding to Bik1. Frame 3: The unoccupied motor domain (in this case, orange) bound to ATP sequesters a free tubulin dimer in its curved conformation. Frame 4: The motor domain facilitates the incorporation of the newly sequestered dimer. The motor domain ahead (orange) hydrolyzes ATP quickly, allowing the other motor domain (blue) to be released from the microtubule protofilament and sequester the next free tubulin dimer. The Bik1-dependent retention enables Kip2 to perform multiple rounds of tubulin incorporation. Frame 5: While this Kip2 molecule is eventually released from the plus-end, such as by phosphorylation on the N-terminus, new Kip2 motors arrive with Bik1 along the lattice. In the absence of Bik1, Kip2 leaves the plus-end as soon as it arrives. Thus, Kip2 fails to promote microtubule growth. The vertical dashed line (black) indicates the length of the microtubule protofilament before Kip2 incorporated tubulin dimers.

both the minus and the plus-ends of microtubules, full-length Kip2-G374A interacts with Bik1 only at SPBs. Thus, the full-length protein cannot bind Bik1 elsewhere than at SPBs when it cannot hydrolyze ATP, and walk to the plus-end. We suggest that domains of the full-length protein, such as its N-terminal extension and its phosphorylation cycle, are involved in this regulation, imposing a strict requirement for Kip2 to pass first through the SPB and microtubule minus-end in order to bind Bik1 and become active as a polymerase at the plus-end (Hibbel et al., 2015).

Proper function of Bik1 requires its CAP-Gly domain, which mediates its direct binding to α-tubulin, EB1/Bim1, and to itself (Akhmanova and Steinmetz, 2008). Indeed, a single point mutation within the CAP-Gly domain of Bik1 (Lys46Glu) that disrupts its interaction with EEY/F motifs was sufficient to nearly abolish the accumulation of Bik1 at microtubule plus-ends and to inactivate the function of Bik1 in microtubule polymerization (Stangier et al., 2018). Similarly, in mammalian cells CLIP-170 promotes microtubule rescue and this requires its two CAP-Gly domains (Komarova et al., 2002). The conserved requirement of

CAP-Gly domains between yeast and mammalian CLIP-170 implies similarity in their mechanisms of action. However, how CAP-Gly domains mediate the function of CLIP family proteins in vivo is still an open question. Our work suggests that CLIP-170 family proteins might act through facilitating the retention of microtubule regulating factors at plus-ends in a CAP-Gly-domain dependent manner. We postulate that the recent notion that both Bik1 and CLIP-170 can phase separate in vitro and assemble into a condensate at microtubule tips in vivo (Ijavi et al., 2021; Miesch et al., 2021; Meier et al., 2022) may provide an important insight for how linker proteins fulfill their very diverse functions in regulating microtubule behavior.

## Materials and methods
### Yeast strains
Yeast strains used in this study are listed in Table S4. All strains are isogenic to S288C. Fluorescent or HA-tagged proteins were tagged at endogenous loci (Knop et al., 1999). All gene deletions were created using the PCR-based integration system (Janke

et al., 2004) and gene deletions were verified by PCR analysis. Specific Kip2 mutations were introduced on a pRS314-Kip2-3xsfGFP:KanMX plasmid or a pRS304-Kip2 plasmid via site-directed mutagenesis (pfu-Turbo, Stratagene). KIP2 locus was then amplified and integrated in a kip2Δ strain and the correct integration was verified by PCR and sequencing.

### Media and growth conditions

Cells were cultured in YEPD (yeast extract peptone, 2% dextrose) for collecting Western blotting samples. For live cell imaging, overnight cultures in SC (synthetic medium, 2% dextrose) were diluted to $OD_{600}$ 0.15 and cultivated for 4 more hours before being placed on an SC-medium agar patch for microscopy imaging.

### Confocal microscopy

A Nipkow spinning disk (Carl Zeiss) equipped with an incubator for temperature was employed. Time-lapse movies were acquired using a back-illuminated EM-CCD camera Evolve 512 (Photometrics, Inc) mounted on the spinning disk microscope with a motorized piezo stage (ASI MS-2000) and 100 × 1.46 NA alpha Plan Apochromat oil immersion objective, driven by Metamorph-based software VisiVIEW (Visitron Systems). 17 Z-section images separated by 0.24 µm increments were captured with the exposure time of 30 ms each; the whole stack took 1.07 s. For imaging aMT dynamics, 80 continuous repetitions were taken. For imaging strains with both GFP and mCherry signals, the GFP channel was always set to 30 ms exposure time and the mCherry channel to 50 ms exposure time. Images in figures represent sum fluorescence intensities across Z-projections. Scale bars represent 2 µm.

### Image and data analysis

Preanaphase cells were collected based on the shape of cells and the size of spindles. For analyzing the profiles of GFP fusion proteins along aMTs, the sum intensity projection of the images was used. A 5-pixel (666.7 nm) width line was used to scan aMTs from plus-ends toward SPBs both in the GFP and the mCherry channels using Fiji (Schindelin et al., 2012), and exported to CSV files. These profiles were then aggregated for further analysis using MATLAB (R2018a; Mathworks), and peak detection for the GFP and mCherry signals was performed, respectively. Profile length was defined as the peak-to-peak distance, and the profiles were then binned into length bins as detailed in the figure legends.

For the purpose of demonstration and for calculating the speeds of fluorescent speckles, kymographs were generated and analyzed using Fiji. Shortly, a 5-pixel width line was placed along preanaphase cytoplasmic microtubules from SPBs toward plus-ends, and kymographs were created using the "Reslice" function without interpolation. The position of the 5-pixel line was adjusted to cover the whole microtubule over time as well as possible. Due to the pivoting of cytoplasmic microtubules, these kymographs do not capture all fluorescent speckles from their origination to dissociation. When the microtubule moves out of the covered area, the corresponding speckles disappear from the kymograph. Conversely, when a part of the microtubule moves into the center of the covered area, dim speckles become brighter. The speed of fluorescent speckles moving toward microtubule plus-ends was calculated by extracting the coordinates of the starting and terminal positions of each speckle using the kymographs.

For fluorescence intensity, a region of interest (ROI) was drawn around the area of interest (AOI) and the integrated density was extracted. An identically sized ROI was put next to the AOI to determine the background signal. The background intensity was subtracted from the ROI intensity to yield the fluorescence intensity (a.u.). For every experiment that was performed for quantification of fluorescence intensity (a.u.), corresponding wild-type cells were imaged and analyzed for comparison to mutant cells. Average values of wild-type cells of different experiments were used for normalization and comparison between experiments.

To determine the length of aMTs, endogenously expressed Bik1-3xGFP or Kip2-3xsfGFP and Spc72-GFP were used as the plus- and minus-end marker, respectively. 3D coordinates of microtubule plus-ends and the corresponding SPBs were extracted with the Low Light Tracking Tool (Krull et al., 2014). The tracking tool does make mistakes when microtubules depolymerize with a very high rate, or when microtubules pivot quickly with large angles, or when the intensity of the plus-end marker is low. Therefore, all tracked trajectories were inspected by eye to find and to correct those very rare mistakes. All of the time series tracking results were analyzed with custom functions written in Matlab (MathWorks). The distance between the b- and m- SPBs is the spindle length. Cells with spindles longer than 2 µm were excluded. The distance between a microtubule plus-end and the corresponding SPB represents the length of the microtubule. Only microtubules longer than 5 pixels (666.7 nm) were considered detectable due to the limit of the microscope resolution. Using this criterion, the maximum length (one for each microtubule) and lifetime of each microtubule within the recorded time window (85.6 s) were extracted. Microtubule growth and shrinkage phases were annotated manually and recorded in Matlab, and the speeds of microtubule growth and shrinkage were calculated using these annotations. Among all mutants analyzed in this work, we observed some long and bending aMTs in Kip2-ΔN cells. We excluded those cells from our analysis. No bending aMTs were observed in other mutants.

### Western blot

For protein extraction, 2 $OD_{600}$ log phase cell cultures were spun down and pellets were washed once with ice-cold PBS, then lysed with Zirconia-Silicate beads in lysis buffer (50 mM Tris, pH 7.5, 150 mM NaCl, 0.5 mM EDTA, 1 mM $MgCl_2$, Roche Complete Protease and phosphatase inhibitors and 0.2% NP-40) on a FastPrep-24 homogenizer. Lysate was cleared by centrifugation at 5,000 × g, 4°C for 5 min. Samples were separated on a 6% SDS-polyacrylamide gel (SDS-PAGE), wet-transferred onto a polyvinylidene fluoride membrane for Western blotting. Antibodies used were primary antibody anti-HA (1:1,000, mouse monoclonal clone 16B12, #MMS-101R; Covance Inc.) and

secondary antibody goat anti-Mouse IgG conjugated to horse-radish peroxidase (1:5,000, #1706516; Bio-Rad).

## Statistics

Each experiment was repeated with three or more independent clones (biological replicates). For wild-type strains, extra technical replicates were performed. The SD is shown in the graphs, or as indicated. n.s. (not significant) or asterisks indicate P values from Student's *t* test, one-way ANOVA, or Kolmogorov-Smirnov test as indicated. Statistical analyses were performed on the means of the replicates.

## Kip2-MD homology model

The homology model of the Kip2-MD was generated in PHYRE2 (Kelley et al., 2015) using the crystal structure of the motor domain of CENP-E (PDB: 1T5C). In the Kip2-MD, the three positively charged patches that represent potential tubulin/microtubule-binding interfaces were identified by a combination of a sequence alignment of the Kip2-MD with other motor domains (human CENP-E, human Kif2C, and human Kif5a) and a structural alignment of the Kip2-MD with the structure of a kinesin-13 motor domain bound to tubulin-DARPin1 (PDB: 5MIO).

## Protein preparations

The DNA encoding the motor domain of *S. cerevisiae* Kip2 (Kip2-MD residues 100–503; UniProt ID: P28743), the non-motor domain of *S. cerevisiae* Kip2 (Kip2-NMD residues 504–706; UniProt ID: P28743), and a truncated version of the non-motor domain of *S. cerevisiae* Kip2 (Kip2-NMD-ΔT residues 504–645; UniProt ID: P28743) were cloned into the pET-based bacterial expression vector PSPCm2, which encodes for an N-terminal 6× His-tag and a PreScission cleavage site using a positive selection method (Olieric et al., 2010). The vector for the expression of the Bik1-CC was already available from our previous work (Stangier et al., 2018).

The production of the proteins was performed in the *Escherichia coli* strain BL21 (DE3; Stratagene) in Luria-Bertani media containing 50 μg/ml of kanamycin. After the cultures had reached an $OD_{600}$ of 0.6 at 37°C, they were cooled down to 20°C, induced with 1 mM isopropyl 1-thio-β-D-galactopyranoside and shaken for another 16 h at 20°C. Then, the cells were harvested and washed with Dulbecco PBS buffer (Millipore). Next, they were sonicated in the presence of the protease inhibitor cOmplete cocktail (Roche) in lysis buffer (50 mM Hepes, pH 8.0, supplemented with 500 mM NaCl, 10 mM imidazole, 2 mM β-mercaptoethanol, 0.1% bovine deoxyribonuclease I).

Proteins were purified by immobilized metal-affinity chromatography (IMAC) on a HisTrap HP Ni$^{2+}$-Sepharose column (GE Healthcare) at 4°C according to the instructions of the manufacturer. The column was equilibrated in IMAC buffer A (50 mM Hepes, pH 8.0, supplemented with 500 mM NaCl, 10 mM imidazole, 2 mM β-mercaptoethanol). Proteins were eluted by IMAC buffer B (IMAC buffer A containing 400 mM imidazole in total). In the case of the Kip2-NMD and Kip2-NMD-ΔT, the N-terminal 6× His-tag was cleaved off by an in-house produced HRV 3C protease (Cordingley et al., 1990) in IMAC

buffer A for 16 h at 4°C. The cleaved samples were reapplied on the IMAC column to separate cleaved from uncleaved protein.

Protein samples were concentrated and loaded onto a SEC HiLoad Superdex 200 16/60 column (GE Healthcare), which was equilibrated in SEC buffer (20 mM Tris-HCl, pH 7.5, supplemented with 150 mM NaCl and 1 mM DTT). The fractions of the respective main peaks were pooled and concentrated to 5–20 mg/ml. Protein quality and identity were analyzed by SDS-PAGE and mass spectrometry, respectively.

All constructs used in in vitro studies are listed in Table S3.

## SEC-MALS

For the SEC-MALS experiment at 25°C, a Superdex200 10/300 or a Superdex75 10/300 (GE Healthcare) column was equilibrated in Tris (20 mM Tris-HCl, pH 7.5, supplemented with 150 mM NaCl and 1 mM DTT; Bik1-CC, Kip2-NMD, and Kip2-NMD-ΔT) or Hepes buffers (20 mM Hepes, pH 7.4, supplemented with 200 mM NaCl and 1 mM DTT; MBP-Kip2-MD, MBP-Kip2-MD-P1, and tubulin) at a flow rate of 0.5 ml/min on an Agilent UltiMate3000 HPLC. 30 μl of the respective single protein or complex was injected onto the column, and the mass was determined using the miniDAWN TREOS and Optilab T-rEX refractive index detectors (Wyatt Technology). For the Bik1-CC, Kip2-NMD, Kip2-NMD-ΔT, MBP-Kip2(71–560)-mCherry, MBP-Kip2(71–560)-P1⁻-mCherry, and tubulin 4–8 mg/ml were applied. In case of the experiments with mixtures of proteins, the individual concentrations of the components were maintained. The Zimm model was chosen for data fitting that was performed in the ASTRA 6 software.

## Microtubule-pelleting assay

Microtubule-pelleting assays of purified proteins were performed as previously described (Campbell and Slep, 2011). Briefly, 10 mg/ml bovine brain tubulin was diluted in a 1 × BRB80 buffer (80 mM K-PIPES, pH 6.8, supplemented, 1 mM EGTA, 1 mM $MgCl_2$, and 1 mM DTT) to 2 mg/ml. After the addition of 0.5 mM GTP, the sample was incubated on ice for 5 min. Microtubule polymerization was started by transfer to 37°C. After 10 min, 0.1, 1, and 10 μM paclitaxel was added stepwise with incubation times of 5 min each. As control, taxol-stabilized microtubules or the respective protein was applied alone. Samples were applied onto a taxol–glycerol cushion that contained 55% 2 × BRB80, 44% glycerol, and 6% 2 mM paclitaxel. After centrifugation at 174,500 *g* for 30 min at 25°C, an aliquot was taken from the supernatant. After removal of the supernatant, the pellet was resuspended in SDS sample buffer. Samples were loaded and analyzed on Coomassie-stained 12% SDS-PAGE gels.

## Mathematical estimation supporting information

Mathematical estimation was performed as previously described (Chen et al., 2019b), except that: the estimation procedure was additionally fit to *bik1Δ* data and *Kip2-ΔT* data, and no fluorescence normalization was required or performed since all data was acquired on January 30, 2018. Statistical significance of the differences in parameter values were computed by sampling from the likelihood (with 2e4 samples), as previously described (Chen et al., 2019b).

## Proteins purification for total internal reflection fluorescence (TIRF) experiments

Tubulin was isolated from pig brain by two cycles of microtubules polymerization/depolymerization in high molarity pipes buffer as described in Castoldi and Popov (2003).

MBP-Kip2-(1-560)WT-RFP and MBP-Kip2-(1-560)P1⁻-RFP constructs were cloned in pET(24)d plasmid and expressed in *E. coli* (Rosetta strain). Bacteria were grown in 2 YT medium at 37°C until culture reached OD 0.6 and then the recombinant protein expression was triggered by the addition of 200 μM 1-thio-β-D-galactopyranoside for 16 h at 16°C. Cells were harvested and resuspended in purification buffer for Kip2-RFP (2XT buffer: 40 mM Tris, 400 mM KCl, 2 mM MgCl2, 1 mM ATP, pH 7.4) or for mCherry-Bik1 (2XH buffer: 40 mM Hepes, 400 mM KCl, 2 mM MgCl2, 1 mM ATP, pH 7.4) with 5 mM β-mercaptoethanol and protease inhibitor (SigmaFAST, protease inhibitor Cocktail EDTA-Free, S8830) and lysed by sonication, and the protein purified using Amylose resin (E8021; NEB) and eluted with 10 mM maltose. The proteins were aliquoted, flash frozen in liquid nitrogen, and stored at –80°C.

## TIRF microscopy assay

Tubulin was labeled with NHS-Biotin (Ez-link NHS-LC-LC-Biotin, 21343; Thermo Fisher Scientific), NHS-ATTO-488 (AD488-31; ATTO-TEC), NHS-ATTO-647 (AD647-31; ATTO-TEC) during a cycle of polymerization/depolymerization as described in (Hyman et al., 1991) MT seeds were polymerized with 20 μM tubulin (with 5% of tubulin-atto647 and 5% tubulin-biotin) with 1 mM Guanosine-5'-[(α,β)-methyleno]triphosphate (GMPCPP) in BRB80 at 37°C for 30 min, centrifuged and resuspended in BRB80. Glass coverslips were cleaned and silanized as described in Brouhard et al. (2008). The flow cells were assembled from the silanized coverslip and a coverglass using double-sided tape (LIMA, 70pc). The flow cell was treated with 50 μg/ml of neutravidin in PBS1x then washed with a solution of 2% Pluronic F-127. The flow cell was then washed with BRB80 and MT seeds were allowed to attach to the neutravidin-coated surface. Then the polymerization mix containing the 7 μM tubulin (supplemented with 5% tubulin-Atto488) and 1 mM GTP in BRB20 buffer (20 mM PIPES, 1 mM MgCl2, 1 mM EGTA, pH 6.9) supplemented with 100 mM KCl, 20 mM glucose, 20 μg/ml glucose oxidase, 8 μg/ml catalase, 0.1 mg/ml BSA, 1 mM DTT, 1 mM GTP, and 1 mM ATP, was flowed in. Increasing concentration of MBP-Kip2-(1-560)WT-RFP and MBP-Kip2-(1-560)P1-RFP (2, 4, and 8 nM) were added to test the concentration dependent effect of the kinesins on MT polymerization in vitro. The MTs polymerization in the different conditions was recorded for 10 min at 32°C using a TIRF microscope (using an objective-based azimuthal ilas2 TIRF microscope, Nikon Eclipse Ti, modified by Roper Scientific and an Evolve 512 camera from Photometrics). Time-lapse recording every 5 s for 10 min at 488, 565 and 647 nm was performed using Metamorph software. The analysis of MTs dynamics and the generation of kymographs were done using ImageJ software.

## Kip2 motility assay in vitro

MTs stabilized with GMPCPP were polymerized with 20 μM tubulin (with 5% of tubulin-atto488 and 5% tubulin-biotin) with 0.5 mM GMPCPP in BRB80 at 37°C for 1 h. Then the GMPCPP-stabilized MTs were attached into the flow chamber functionalized with neutravidin and passivated with pluronic-F-127 as described above. Then 1 nM of MBP-Kip2-(1-560)WT-RFP and MBP-Kip2-(1-560)P1-RFP in BRB20 buffer (20 mM PIPES, 1 mM MgCl2, 1 mM EGTA, pH 6.9) supplemented with 100 mM KCl, 20 mM glucose, 20 μg/ml glucose oxidase, 8 μg/ml catalase, 0.1 mg/ml BSA, 1 mM DTT, 1 mM GTP, and 1 mM ATP, was flowed in the flow chamber. Time-lapse recording every 5 s for 10 min at 488 and 565 nm was performed using Metamorph software. The analysis of Kip2 motility and the generation of kymographs were done using ImageJ software.

## Online supplemental material

Fig. S1 shows representative 3D aMT length tracking results and the distribution of Kip2 mutants along microtubules. Fig. S2 shows Kip2-Bik1 interaction is not required for efficient targeting of Kip2, but of Bik1, to microtubule plus-ends. Fig. S3 shows experimental and estimate results of Kip2 distribution along microtubules. Fig. S4 shows details of tubulin bound Kip2 motor domain homology model and effects of indicated mutants in heterozygous yeast cells. Fig. S5 shows Kip2 residues K294 and R296 are dispensable for Kip2 motility in vivo and in vitro. Video1 is a representative time-lapse movie showing Kip3-Kip2-chimera molecules accumulate on microtubule plus-ends. Video 2 shows Kip2-NMD-mNeonGreen accumulates on both the growing and shrinking cytoplasmic microtubule plus-ends. Data S1 shows quantification of maximum aMT length (μm) in metaphase cells of indicated genotype for Fig. 1 F and Fig. 5 H; quantification of aMT dynamics in wildtype and kip2Δ preanaphase cells at 30°C for Fig. 1, G–I; normalized GFP fluorescence intensity (%) on aMT plus-ends for Fig. 4 B; ratio of tubulin intensity over Kip2 or MBP intensity from free GDP-tubulin co-IP experiment for Fig. 6, C–E and Table S2; quantification of the speeds (μm/min) of GFP speckles moving along aMTs in vivo for Fig. S5, B and D; quantification of the speeds (μm/min) of GFP speckles moving along preanaphase cytoplasmic microtubules for Fig. S2 B; relative Bik1-3xGFP fluorescence (%) associated with bud-directed cytoplasmic microtubules in cells of indicated genotype for Fig. S2 C; and relative Kip2-G374A-3xsfGFP fluorescence (%) associated with bud-directed SPBs in wildtype and bik1Δ cells for Fig. S2 F. Table S1 shows quantification of preanaphase astral microtubule length and dynamics in living cells. Table S2 shows quantification of microtubule length and dynamics in vitro with 7 μM porcine tubulin. Table S3 shows all recombinant proteins used in in vitro studies. Table S4 shows yeast strains used in this study.

## Data availability

All raw image data and code are available in the main text, the supplementary materials, or at https://gitlab.com/csb.ethz/kip2-bik1-manuscript/.

## Acknowledgments

We acknowledge financial support by the SystemsX.ch RTD Grant #2012/192 TubeX to Y. Barral, M.O. Steinmetz, and J. Stelling.

M.O. Steinmetz is supported by the Swiss National Science Foundation grant 310030_192566. M.P. Czub has received funding from the European Union's Horizon 2020 research and innovation program under the Marie Skłodowska-Curie grant agreement No 884104 (PSI-FELLOW-III-3i). D. Liakopoulos acknowledges the support of the French Agence Nationale de la Recherche, grant ANR-14-CE09-0014-01 (ReconstMT-Act) and the imaging facility MRI, member of the national infrastructure France-BioImaging supported by the French National Research Agency (ANR-10-INBS-04, «Investments for the future»).

Author contributions: X. Chen: conceptualization, data curation, formal analysis, investigation, methodology, resources, validation, visualization, writing—original draft, writing—review & editing. L.A. Widmer: conceptualization, methodology, software, formal analysis, data curation, visualization, writing—review & editing. M.M. Stangier: conceptualization, data curation, formal analysis, investigation, validation, methodology, writing—review & editing. M.P. Czub: data curation, formal analysis, investigation, validation, methodology, writing—review & editing. D. Liakopoulos: conceptualization, investigation, resources, writing—review & editing, supervision, project administration, funding acquisition. D. Portan: conceptualization, validation, formal analysis, investigation, resources, data curation, writing—original draft, visualization, supervision. J. Stelling: conceptualization, supervision, funding acquisition, writing—review & editing. M.O. Steinmetz: conceptualization, supervision, funding acquisition, writing—review & editing. Y. Barral: conceptualization, project administration, supervision, funding acquisition, methodology, writing—original draft, writing—review & editing.

Disclosures: L.A. Widmer reported personal fees from Novartis Pharma AG outside the submitted work. No other disclosures were reported.

Submitted: 21 October 2021

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

**Supplemental material**

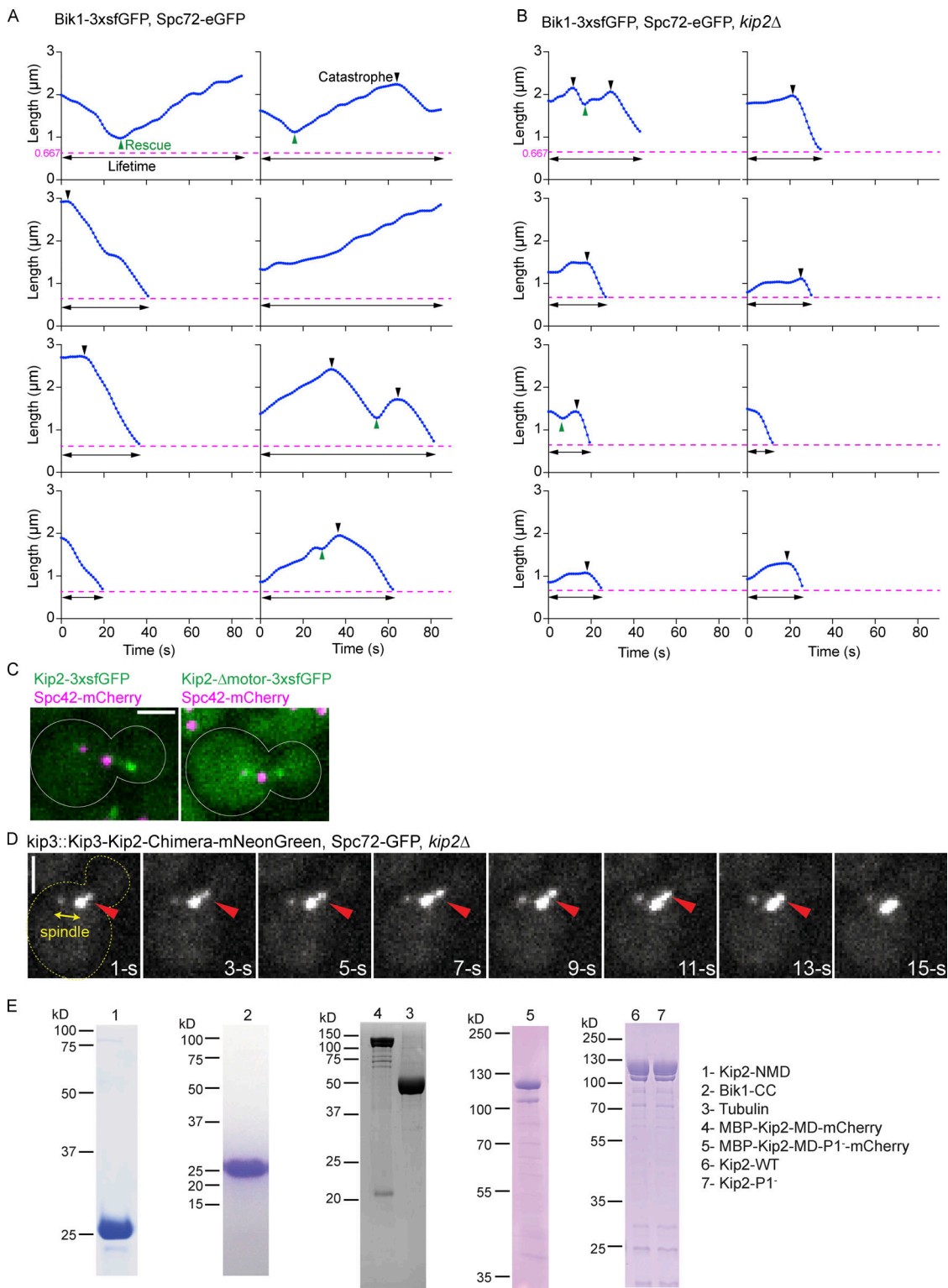

Figure S1. **Representative 3D aMT length tracking results, distribution of Kip2 mutants along microtubules, and SDS-PAGE of proteins used in in vitro assays. (A and B)** Representative 3D aMT length was extracted from control (A) and *kip2Δ* (B) cells using Bik1-3xsfGFP and Spc72-eGFP as microtubule plus- and minus-end markers, respectively. The detection limit (666.7 nm due to the microscope resolution) is marked as a magenta dashed line. Rescue (green) and catastrophe (black) events are marked with triangles. The lifetime of each aMT is indicated with a black line with two arrowheads. **(C)** Representative images of full-length Kip2-3xsfGFP and Kip2-Δmotor-3xsfGFP accumulation on microtubule tips. **(D)** The chimera protein is composed of Kip3-MD and Kip2-NMD, then C-terminally fused with mNeonGreen, expressed from *KIP3* locus. The red arrowhead marks the plus-end of the aMT. SPBs are visualized with Spc72-GFP. Corresponds to Video 1. The time-lapse movie has 1.07 s intervals; only representative time points are shown here. Scale bar, 2 µm. **(E)** Coomassie-Blue-stained SDS-PAGE analysis of indicated proteins. Details of the constructs are summarized in Table S3. The Kip2-NMD image was cropped from the gel shown in Fig. 2 C. Source data are available for this figure: SourceData FS1.

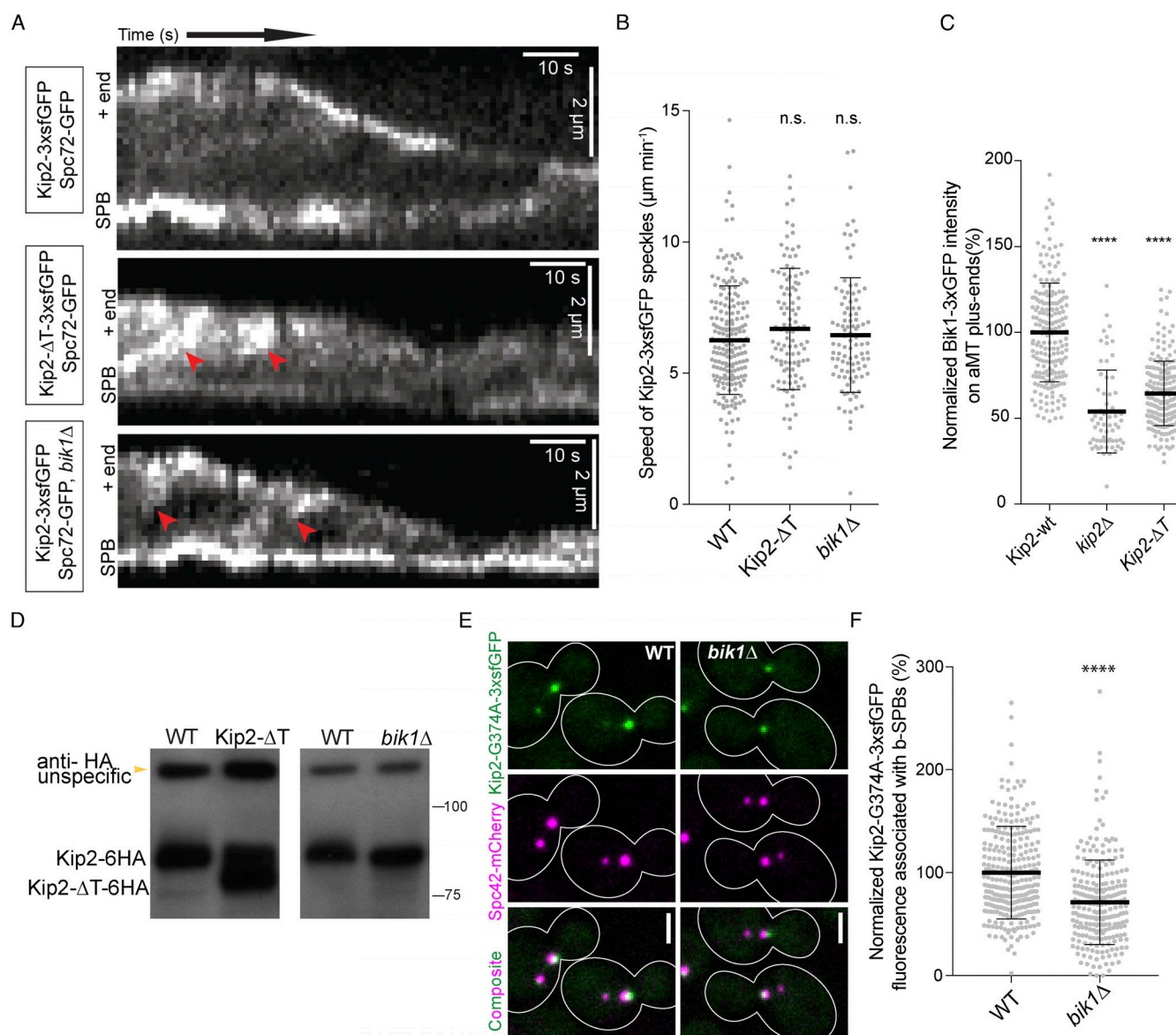

Figure S2. **The Kip2–Bik1 interaction is not required for efficient targeting of Kip2, but of Bik1, to microtubule plus-ends. (A)** Representative kymographs drawn from time-lapse series of preanaphase cells of the indicated genotype. SPBs were visualized with Spc72-GFP. Red arrowheads denote the speckles that appeared along the shaft of microtubules. **(B)** Quantification of Kip2 speckle moving speed using the kymographs shown in A. More than 90 speckles analyzed per condition. **(C)** Measurements of Bik1-3xGFP fluorescence intensity (%) on cytoplasmic microtubule plus-ends in cells of the indicated genotype. More than 90 cells analyzed per condition. **(D)** Western blot analysis of endogenously expressed Kip2-6HA and Kip2-ΔT-6HA. Lysates were prepared from cycling cells of the indicated genotype. **(E)** Representative images of preanaphase cells expressing the SPB marker Spc42-mCherry (magenta) and the ATPase deficient variant Kip2-G374A-3xsfGFP (green) in the presence and absence of Bik1. **(F)** Measurements of Kip2-G374A-3xsfGFP fluorescence intensity (%) associated with b-SPBs in cells shown in E. More than 90 speckles cells per condition. Statistical significance was calculated using two-tailed Student's *t* test. **** P < 0.0001; n.s., not significant. Source data for B, C, and F are available in Data S1. Source data are available for this figure: SourceData FS2.

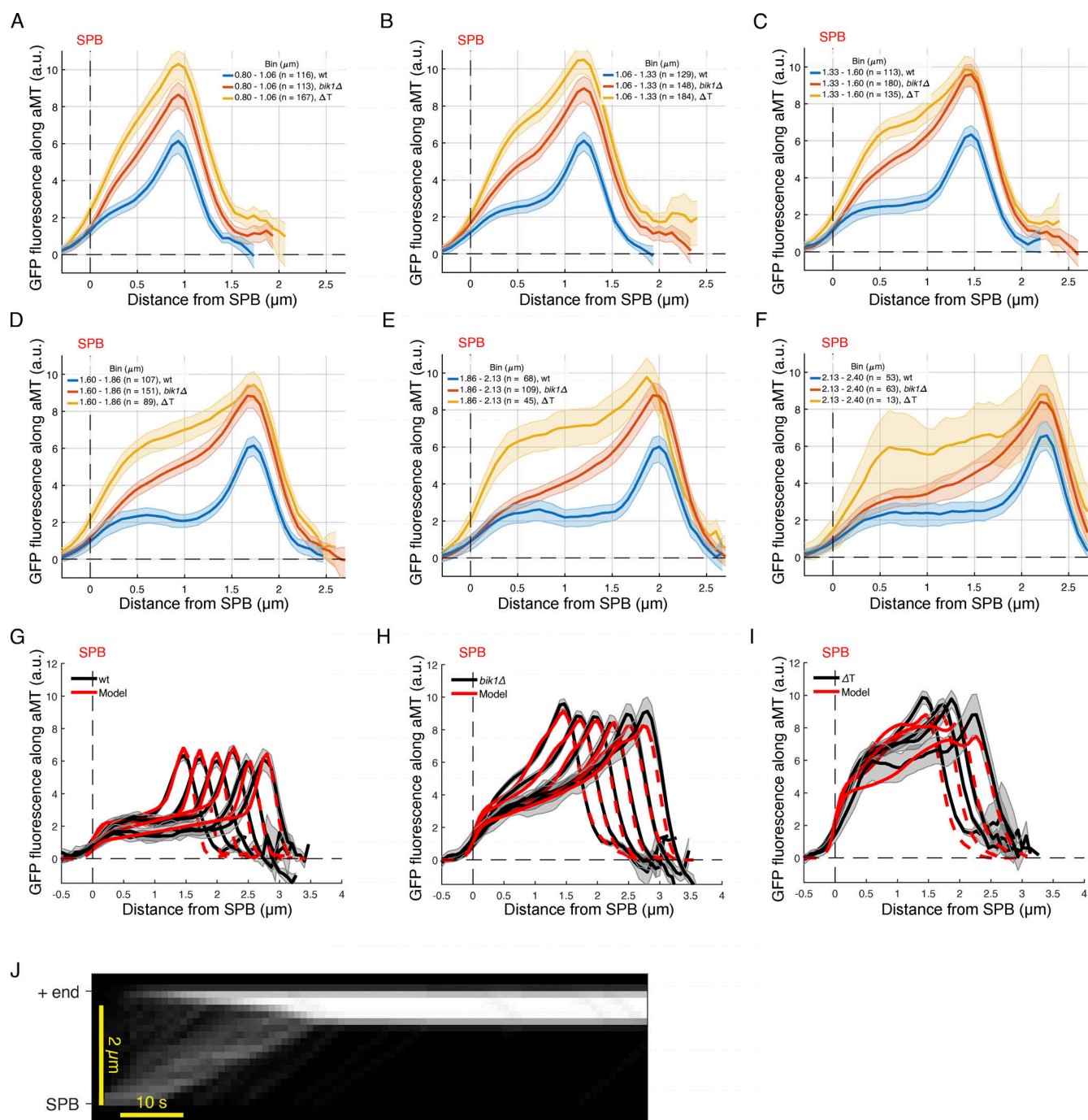

Figure S3. **Experimental and estimate results of Kip2 distribution along microtubules. (A–F)** Quantifications of fluorescence intensities (a.u.) from endogenous Kip2-ΔT-3xsfGFP and Kip2-3xsfGFP in the presence or absence of Bik1 along preanaphase cytoplasmic microtubules binned by microtubule length: (A) 0.80–1.06 µm, (B) 1.06–1.33 µm, (C) 1.33–1.60 µm, (D) 1.60–1.86 µm, (E) 1.86–2.13 µm, (F) 2.13–2.40 µm. **(G–I)** Experimental Kip2-ΔT-3xsfGFP (H), Kip2-3xsfGFP in the presence (G) or absence (I) of Bik1 fluorescence (a.u.) mean profile (black) and standard error (gray) with respective mean in silico estimation fits (red) for microtubules binned by length. Red dashed lines past plus-end and SPB indicates estimation extrapolations without support by data. **(J)** In silico kymograph with on rate constant of 6.1E-4 (nM Kip2)$^{-1}$ s$^{-1}$, in rate constant of 3.1E-1 (nM Kip2)$^{-1}$ s$^{-1}$, off rate constant of 2.3E-2 s$^{-1}$, and total Kip2 concentration of 35 nM.

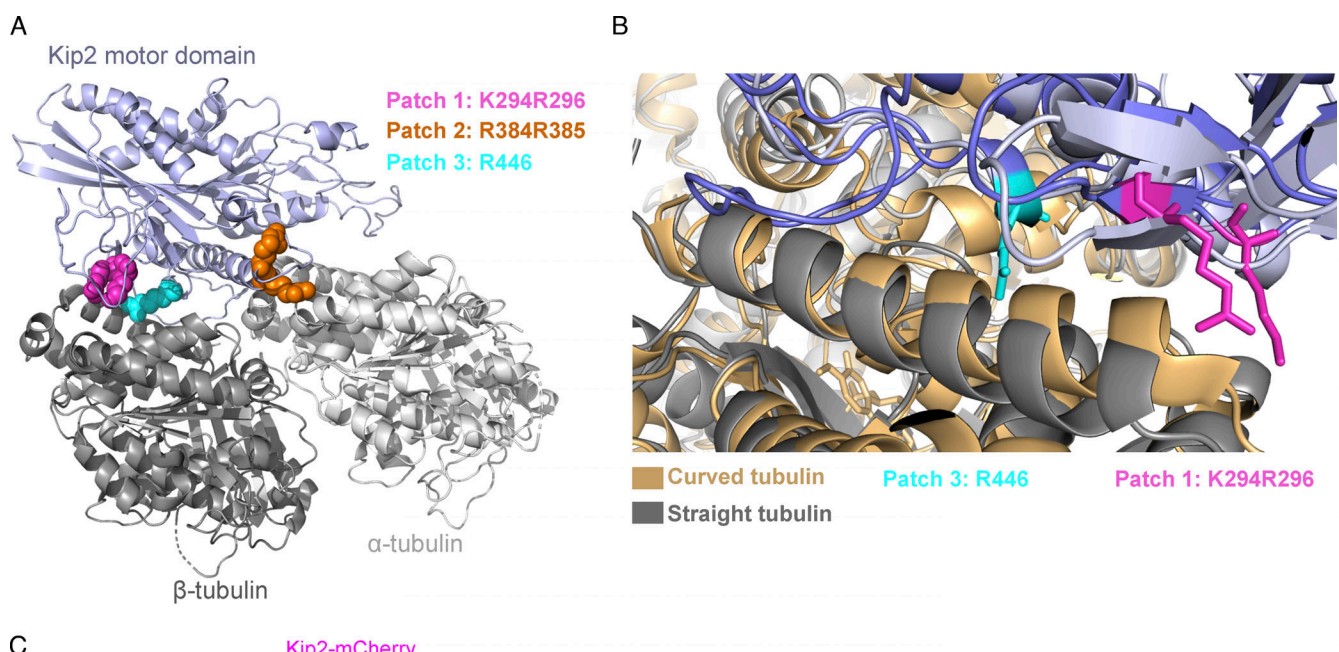

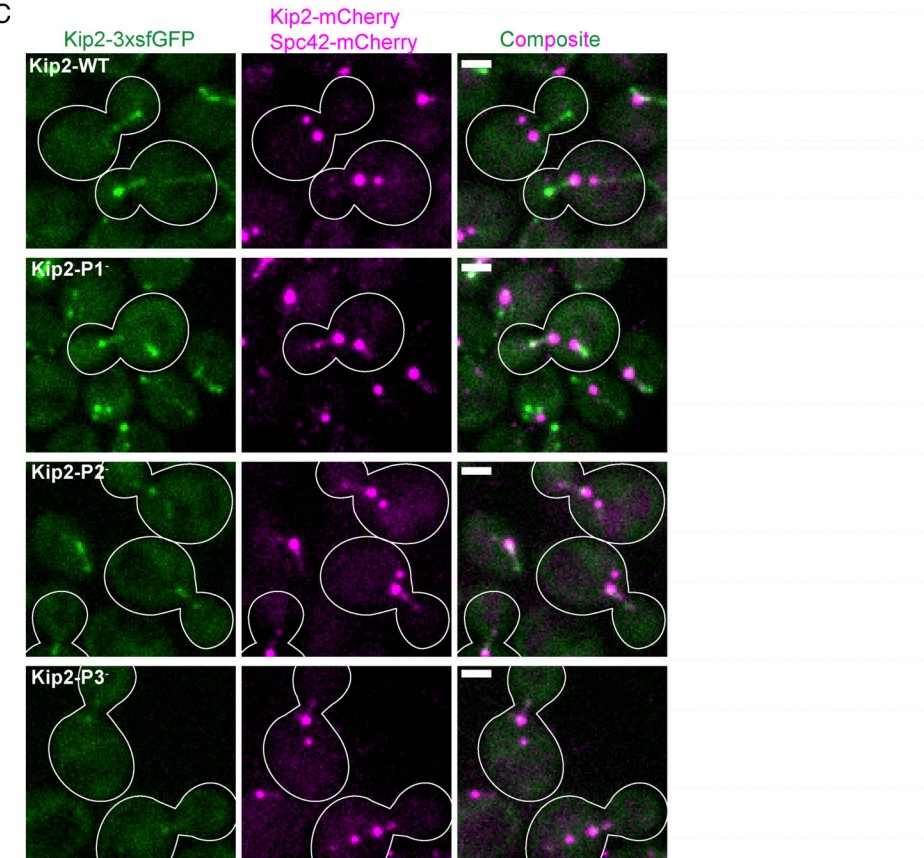

Figure S4. **Details of tubulin-bound Kip2-MD homology model and effects of indicated mutants in heterozygous yeast cells. (A)** Kip2-MD homology model (blue) in complex with a αβ-tubulin (gray) heterodimer. The three positively charged Kip2-MD surface patches that are crucial for tubulin or microtubule binding are highlighted in different colors (see corresponding legend). **(B)** Predicted binding of Kip2-MD (blue) to the curved (brown) and straight (gray) conformational states of tubulin. Residues K294 and R296 (P1, purple) and R446 (P3, sky blue) are highlighted to illustrate that P1 may selectively bind to the curved conformation of tubulin. **(C)** Localization of the wild-type and the tubulin-binding patch mutants of Kip2 in heterozygous diploid cells. Representative images of preanaphase heterozygous diploid cells expressing wild-type and tubulin-binding patch mutants of Kip2 fused to 3xsfGFP (green), together with wild-type Kip2 fused to mCherry. SPBs are visualized with Spc42-mCherry (magenta). Scale bars, 2 μm.

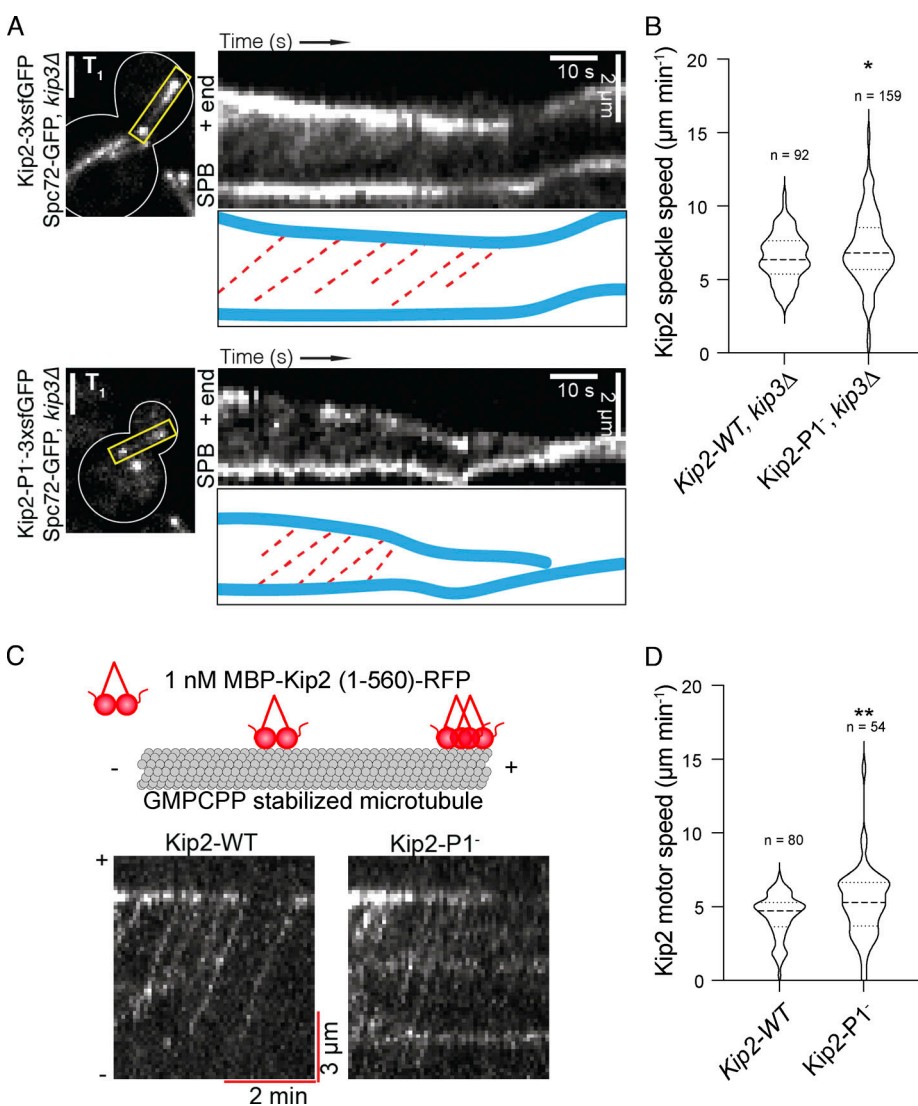

Figure S5. **Kip2 residues K294 and R296 are dispensable for Kip2 motility in vivo and in vitro. (A)** Representative preanaphase cells ($T_1$: the first frame), kymographs drawn from time-lapse series, and highlighted trajectories (red dashed lines) of Kip2-3xsfGFP (top) and Kip2-P1$^-$-3xsfGFP (bottom) speckles moving along a metaphase aMT. Spc72-GFP was used to visualize SPBs. *KIP3* was deleted to increase the aMT length. Scale bars, 2 μm. **(B)** Quantification of Kip2 speckle moving speed using the kymographs shown in A. More than 90 speckles analyzed per condition. **(C)** Representative kymographs showing random landing along GMPCPP-stabilized microtubule lattices, processive motility, and plus-end accumulation of individual Kip2-WT and Kip2-P1$^-$ motors. **(D)** Quantification of Kip2 motor speed using the kymographs shown in C. More than 55 speckles analyzed per condition. Statistical significance was calculated using two-tailed Student's *t* test; * P < 0.05; ** P < 0.01. Source data for C and D are available in Data S1.

Video 1. **A representative time-lapse movie shows Kip3-Kip2-chimera molecules accumulate on microtubule plus-ends.** Corresponds to Fig. S1 D. The movie consists of 80 frames taken every 1.07 s and the frame speed is sped up by threefold for better visualization. Scale bar, 2 μm.

Video 2. **Kip2-NMD-mNeonGreen accumulates on both the growing and shrinking cytoplasmic microtubule plus-ends.** The red arrowhead marks the plus-end. SPBs are visualized with Spc72-GFP. Corresponds to Fig. 4 C. The movie consists of 80 frames taken every 1.07 s and the frame speed is sped up by threefold for better visualization. Scale bar, 2 μm.

**Provided online are Data S1, Table S1, Table S2, Table S3, and Table S4. Data S1 shows quantification of maximum aMT length (μm) in metaphase cells of indicated genotype for Fig. 1 F and Fig. 5 H; quantification of aMT dynamics in wildtype and kip2Δ preanaphase**

cells at 30℃ for Fig. 1, G–I; normalized GFP fluorescence intensity (%) on aMT plus-ends for Fig. 4 B; ratio of Tubulin intensity over Kip2 or MBP intensity from free GDP-Tubulin co-IP experiment for Fig. 6, C–E and Table S2; quantification of the speeds (µm/min) of GFP speckles moving along aMTs in vivo for Fig. S5, B and D; quantification of the speeds (µm/min) of GFP speckles moving along preanaphase cytoplasmic microtubules for Fig. S2 B; relative Bik1-3xGFP fluorescence (%) associated with bud-directed cytoplasmic microtubules in cells of indicated genotype for Fig. S2 C; and relative Kip2-G374A-3xsfGFP fluorescence (%) associated with bud-directed SPBs in wildtype and bik1∆ cells for Fig. S2 F. Table S1 shows quantification of preanaphase astral microtubule length and dynamics in living cells. Table S2 shows quantification of microtubule length and dynamics in vitro with 7 µM porcine tubulin. Table S3 lists all recombinant proteins used in in vitro studies. Table S4 shows yeast strains used in this study.

