## [Peer Review File · The Journal of Cell Biology]

The motor domain of the kinesin Kip2 promotes microtubule polymerization at microtubule tips

Xiuzhen Chen, Didier Portran, Lukas Widmer, Marcel Stangier, Mateusz Czub, Dimitris Liakopoulos, Joerg Stelling, Michel Steinmetz, and Yves Barral

Corresponding Author(s): Yves Barral, ETH Zurich

Review Timeline:

Submission Date:	2021-10-21
Editorial Decision:	2021-12-13
Revision Received:	2023-01-01
Editorial Decision:	2023-02-21
Revision Received:	2023-03-10

Monitoring Editor: Melissa Gardner

Scientific Editor: Dan Simon

Transaction Report:

DOI: <https://doi.org/10.1083/jcb.202110126>

December 13, 2021

Re: JCB manuscript #202110126

Prof. Yves Barral
ETH Zurich
Biology
Institute of Biochemistry ETH Zurich Schafmattstrasse 18
Otto-Stern-Weg 3
ZUERICH 8093
Switzerland

Dear Prof. Barral,

Thank you for submitting your manuscript entitled "The motor domain of the kinesin Kip2 promotes microtubule polymerization at microtubule tips." The manuscript was assessed by expert reviewers, whose comments are appended to this letter. We invite you to submit a revision if you can address the reviewers' key concerns, as outlined here.

You will see that the reviewers are enthusiastic about your work and ask for additional experiments to provide further supporting evidence for your conclusions and clarify the model. Most crucial is the need for more insight into how Bik1 promotes recruitment of Kip2 to SPBs and to assay how Bik1 affects Kip2 mediated polymerization in vitro. Also necessary are the requests to assay effects of mutations in the Kip2 patches on binding to tubulin and ruling out possible effects of microtubule bending. There are also several technical comments and requests for method clarifications and more replicates for quantifications. These should also be addressed in full. Reviewer #2 suggests experiments to rule out a previously proposed model that Kip2 is directly shuttling tubulin and to investigate whether Bik1 prolongs the residence time of Kip2 at ends. We agree that these would be very interesting and would significantly increase the paper's impact so we encourage you to do these if possible. However, this is not required for revision.

GENERAL GUIDELINES:

Text limits: Character count for an Article is < 40,000, not including spaces. Count includes title page, abstract, introduction, results, discussion, acknowledgments, and figure legends. Count does not include materials and methods, references, tables, or supplemental legends.

Figures: Articles may have up to 10 main text figures. Figures must be prepared according to the policies outlined in our Instructions to Authors, under Data Presentation, <https://jcb.rupress.org/site/misc/ifora.xhtml>. Scale bars must be present on all microscopy images, including inset magnifications. Molecular weight or nucleic acid size markers must be included on all gel electrophoresis. All figures in accepted manuscripts will be screened prior to publication.

References: There is no limit to the number of references cited in a manuscript. References should be cited parenthetically in the text by author and year of publication. Abbreviate the names of journals according to PubMed. Please check your reference list for duplicates, we've noticed several duplications (16&25, 17&22, 15&24, 21&44, 6&45).

Statistical analysis: Error bars on graphic representations of numerical data must be clearly described in the figure legend. The number of independent data points (n) represented in a graph must be indicated in the legend. Statistical methods should be explained in full in the materials and methods. For figures presenting pooled data the statistical measure should be defined in the figure legends. Please also be sure to indicate the statistical tests used in each of your experiments (both in the figure legend itself and in a separate methods section) as well as the parameters of the test (for example, if you ran a t-test, please indicate if it was one- or two-sided, etc.). Also, if you used parametric tests, please indicate if the data distribution was tested for normality (and if so, how). If not, you must state something to the effect that "Data distribution was assumed to be normal but this was not formally tested."

Materials and methods: Should be comprehensive and not simply reference a previous publication for details on how an experiment was performed. Please provide full descriptions (at least in brief) in the text for readers who may not have access to referenced manuscripts. The text should not refer to methods "...as previously described."

1) For all cell lines, vectors, constructs/cDNAs, etc. - all genetic material: please include database / vendor ID (e.g., Addgene, ATCC, etc.) or if unavailable, please briefly describe their basic genetic features, even if described in other published work or

gifted to you by other investigators (and provide references where appropriate). Please be sure to provide the sequences for all of your oligos: primers, si/shRNA, RNAi, gRNAs, etc. in the materials and methods. You must also indicate in the methods the source, species, and catalog numbers/vendor identifiers (where appropriate) for all of your antibodies, including secondary. If antibodies are not commercial please add a reference citation if possible.

2) Microscope image acquisition: The following information must be provided about the acquisition and processing of images:

- a. Make and model of microscope
- b. Type, magnification, and numerical aperture of the objective lenses
- c. Temperature
- d. Imaging medium
- e. Fluorochromes
- f. Camera make and model
- g. Acquisition software
- h. Any software used for image processing subsequent to data acquisition. Please include details and types of operations involved (e.g., type of deconvolution, 3D reconstitutions, surface or volume rendering, gamma adjustments, etc.).

IMPORTANT: It is JCB policy that if requested, original data images must be made available. Failure to provide original images upon request will result in unavoidable delays in publication. Please ensure that you have access to all original microscopy and blot data images before submitting your revision.

Supplemental information: There are strict limits on the allowable amount of supplemental data. Articles may have up to 5 supplemental figures. Up to 10 supplemental videos or flash animations are allowed. A summary of all supplemental material should appear at the end of the Materials and methods section.

Please note that JCB now requires authors to submit Source Data used to generate figures containing gels and Western blots with all revised manuscripts. This Source Data consists of fully uncropped and unprocessed images for each gel/blot displayed in the main and supplemental figures. Since your paper includes cropped gel and/or blot images, please be sure to provide one Source Data file for each figure that contains gels and/or blots along with your revised manuscript files. File names for Source Data figures should be alphanumeric without any spaces or special characters (i.e., SourceDataF#, where F# refers to the associated main figure number or SourceDataFS# for those associated with Supplementary figures). The lanes of the gels/blots should be labeled as they are in the associated figure, the place where cropping was applied should be marked (with a box), and molecular weight/size standards should be labeled wherever possible.

As you may know, the typical timeframe for revisions is three to four months. However, we at JCB realize that the implementation of measures that limit spread of COVID-19 also pose challenges to scientific researchers. Therefore, JCB has waived the revision time limit. We recommend that you reach out to the editors once your lab has reopened to decide on an appropriate time frame for resubmission. Please note that papers are generally considered through only one revision cycle, so any revised manuscript will likely be either accepted or rejected.

Thank you for this interesting contribution to Journal of Cell Biology. You can contact us at the journal office with any questions, cellbio@rockefeller.edu or call (212) 327-8588.

Sincerely,

Melissa Gardner, PhD
Monitoring Editor
Journal of Cell Biology

Dan Simon, PhD
Scientific Editor
Journal of Cell Biology

Reviewer #1 (Comments to the Authors (Required)):

The microtubule plus-end directed molecular motor Kip2, in combination with the CLIP-170 homolog Bik1, has been shown to be important for cytoplasmic microtubule dynamics in budding yeast. The molecular mechanisms by which Kip2 promotes tubulin polymerization in cells and the contribution of Bik1 during this process, however, are not fully understood. In the current study, the authors use a combination of live cell imaging with high temporal resolution, in vitro biochemical and biophysical experiments and mathematical modelling to interrogate new aspects of the Kip2-Bik1 system. They show that the Kip2-NMD (non-motor domain) is sufficient for this interaction and that it depends on the c-terminal tail of the Kip2-NMD. Plus-end polymerization activity in vivo requires the motor domain of Kip2 and the authors detect binding to tubulin dimers in vitro. Structural homology models allow the authors to propose three positively charged patches on the motor domain to be selectively involved in microtubule lattice binding, versus binding to tubulin dimers. Interestingly, mutation of patch 1 (P1), allows motility in vitro, plus-end localization in vivo, yet seems to lack polymerization activity. The authors propose a model in which Kip2 promotes the incorporation by binding to free tubulin dimers, once it has reached the plus-end. Bik1 increases the retention of Kip2 at the plus-end to support this effect.

Overall this is an interesting study with a nice combination of in vivo and in vitro analysis. While there has been a fair amount of work on Kip2 already, there are a number of interesting novel aspects in this study.

Major points:

1. A novel aspect of the current study is the identification of the distinct patches P1, P2, P3 on the Kip1 MD involved in the binding of microtubules versus tubulin dimers. I missed the demonstration that these mutants indeed affect tubulin dimer binding in vitro (like in Figure 5A). This seems to me a crucial point in the current study, especially regarding the effect of the P1 mutation.

2. The use of an in vitro reconstitution assay for tubulin polymerization to demonstrate an effect of the Kip2 P1 mutation (Figure 6) is appreciated. Along these lines it is a bit of a pity that the first part of the study, which mapped crucial elements of the Kip2-Bik1 interaction, is not connected to these in vitro experiments in a similar manner. The authors cite that Bik1 is not required for Kip2 mediated polymerization in vitro, but there might be a quantitative effect of Bik1 in the reconstituted system? It seems that more features of the model proposed in Figure 7 could be tested in this setup.

Minor points:

Figure 2 B,D,F: It would be good to show the corresponding SDS PAGE, this could be done in the supplement.

Figure 2E: it would be nice to show raw ITC data as heat curves, this could be done in the supplement.

Figure 3: I found the order of the B and C to be confusing. Better to show the wildtype first?

Figure 5 A: I missed data for Kip2 MD alone and a corresponding analysis by SDS-PAGE of the SEC fractions.

I found the usage of the terms Kip2 NMD and Kip2 tail sometimes confusing. Sometimes NMD and Tail seemed synonymous for the non-motor part, sometimes "tail" only meant the extreme carboxyterminus. The authors should check again for consistency throughout the manuscript.

Page 5: Heading of the section: one should qualify the statement with "astral" microtubule growth in vivo

Reviewer #2 (Comments to the Authors (Required)):

In this manuscript, Chen et al. continue their investigation of the budding yeast kinesin polymerase Kip2, focusing on elucidating the molecular mechanisms underlying its microtubule polymerase activity. They employ a range of methods, from live-cell imaging of budding yeast, through biochemical and structural approaches to probe protein interactions, to TIRF-based reconstitution assays with purified protein components. The authors report that the c-terminal tail of Kip2 mediates interaction with its binding protein Bik1 (yeast Clip-170 ortholog), and that this tail domain is important for the Kip2's polymerase activity in vivo. They postulate that Bik1-Kip2 interaction contributes to Kip2's microtubule plus-end tracking and retention at plus ends in vivo. On the other hand, the authors report that Kip2's motor domain binds soluble tubulin, and that the residues involved in tubulin interaction (K294 & R296) are necessary for Kip2's polymerase activity in vitro. Taken together, the authors propose that Kip2-Bik1 interaction facilitates prolonged residency of Kip2 at the microtubule plus end, where motor domains of Kip2 facilitate microtubule polymerization through a direct interaction with soluble tubulin dimers.

The study is of high quality, and the results are presented in a clear and logical fashion. Several interesting questions and controversies remain. For example, in their previous work (Chen et al. 2019), the authors reported that Kip2 targeting to spindle pole bodies (SPB) depends on Bub2 and Bfa1, while Bik1 primarily targets microtubule plus ends. In this work, however, the authors conclude that Bik1 promotes Kip2 recruitment to SPBs (see Discussion), although it is not completely clear how that would work. Additionally, the conclusion that Bik1 increases Kip2's residency time at microtubule plus ends is not directly demonstrated, and the corresponding conclusions should thus be toned down. Finally, the authors are in a position to directly disprove a competing tubulin-shuttle model (suggested by Hibbel et al. 2015), which would provide significant support to the model proposed in this manuscript.

Specific comments:

Figure 1:

- It would be very insightful to show an example image/kymograph from which microtubule dynamics parameters (lifetime, growth and shrinkage rates) are extracted
- Representation of microtubule growth and shrinkage rates on a log plot (Figure 1G) is rather unintuitive - the authors should instead plot microtubule growth rates and microtubule shrinkage rates separately, both on a linear plot
- Numbers of measurements should be added to the caption for all data (e.g. panels 1F and 1G)

Figure 2:

- Panels are incorrectly labeled.

Figure 3:

- This figure is the most confusing, while at the same time essential for the conclusions on how Kip2-Bik1 interaction encodes Kip2 localization in vivo. Since the intention is to directly compare wt vs delT Kip2 constructs in the presence and absence of Bik1, it would be more useful in 3B-D to plot intensity distributions for different mutants against each other on the same graph (e.g. just take 2.5 um long bin and plot the three distributions together)
- The significance of slopes and intercepts in 3B-D is not discussed.
- While the authors have previously employed and extensively discussed their model (Chen et al. 2019), the significance of the modeling results in 3F-H is not clear and sufficiently elaborated. Modeling results should be overlaid with experimental data for direct comparison, as previously done in Chen et al. 2019. It is unclear to what extent the model predicts a unique set of effects on the modeled parameters. For example, could e.g. obtained differences in in-rate be compensated by differences in on-rate and/or concentration at SPBs? Is the 30% reduction in SPB body localization (established using ATP-deficient mutant) explicitly modeled?
- Results presented in 3G should be plotted separately on a linear (not log) scale for clarity.
- Importantly, while 3H shows that modeling estimates a 2-fold increase in Kip2 out rate in the absence of Bik1 interaction, which corresponds to end-residency being reduced from 0.3 s to ~ 0.15 s, previous in vitro results (Hibbel et al. 2015) demonstrated that Kip2 residency times on plus ends on its own in vitro are two orders of magnitude higher (~30 s). Given such a large discrepancy between in vitro and in vivo results, it is hard to appreciate the potential effect that just a 2-fold modulation of the out rate might have.

Figure 4:

- The data in this figure (especially panel C) would be well supported by accompanying intensity linescans, allowing to directly compare investigated protein distributions/co-localizations.
- It is a little bit confusing that Kip2-NMD is targeted to plus ends via Bik1 interaction, but the full-length ATP-ase deficient Kip2 does not localize to plus ends in the presence of Bik1. Shouldn't this mutant still be able to bind Bik1 through its tail domain (have the authors checked this)? How is this observation explained?

Figure 7:

- The authors propose a model in which Kip2-soluble tubulin dimer interaction is only possible if a motor head is not engaged with the microtubule lattice, restricting it to microtubule plus ends. To further eliminate the possibility that Kip2 is directly shuttling tubulin (as proposed in Hibbel et al. 2015), the authors should perform TIRF reconstitution experiments (like in Figure 6) at much lower Kip2-GFP and labeled tubulin concentrations (i.e. 100 pM-1 nM Kip2 and sub-micromolar tubulin range) which would directly reveal whether Kip2-tubulin interactions can occur along the microtubule lattice, and/or only at the ends.
- The evidence that Bik1 prolongs the residence time of Kip2 at ends is weak, as no direct measurements of residence times have been performed. This, of course, could be done using in vitro TIRF approach with purified Bik1. This may well be outside the scope of the manuscript, but in that case the authors should restrain from strong statements that this is indeed what is going on in cells (see Discussion).

Reviewer #3 (Comments to the Authors (Required)):

The manuscript from Chen et al. addresses the important question of how kinesin proteins regulate microtubule (MT) dynamics

overall, and specifically how the kinesin Kip2 promotes MT polymerization and stabilization. This activity is unclear because Kip2 requires its partner Bik1 to promote MT growth/stability in vivo, yet is capable of this alone in vitro. Bik1 itself is required for MT growth/stability in vivo. This manuscript investigates the mechanism of how Kip2 promotes MT growth and stability. The authors examine MT dynamics in cells and show that Kip2 likely promotes both MT polymerization and inhibits catastrophes. The various Kip2 truncations used are informative for the molecular regions required for MT growth, recruitment to SPBs, interaction with tubulin and with Bik1. The authors also test binding interactions between these regions and tubulin, MTs, and parts of Bik1 (CC domain). The authors find no evidence of binding of Kip2-NMT domain to tubulin or MTs using the construct from bacteria and conditions used. Interestingly, they do find that the CC domain of Bik1 does not interact with the CC of Kip2, but rather mainly via the tail region. Using a combination of in vivo imaging and mathematical modeling the authors present data that Bik1 likely recruits Kip2 to SBPs for transport along MTs, and prevents Kip2 from loading along the MT lattice. Bik1 also seems required for Kip2 remaining associated with MT +ends. Data also shows that the NMD of Kip2 recruits Kip2 to the SBP via Bik1, and that this effect may not occur directly at the +end, hence, Kip2 and Bik1 interactions originate the SPB. The authors identify three regions of positive charge on the MT-binding surface and mutate them, showing that P2 and P3 mutants fail to localize properly, whereas P1 mutants can. Although the P1 mutant localizes to +ends, the MTs are short as in a kip2 null. Thus, the P1 patch is needed to promote MT polymerization in vivo. The authors then test the function of Kip2-MD-P1 compared to WT Kip2-MD on MT polymerization and stability in vitro, with data suggesting that the P1 mutant is inefficient at promoting MT growth and stability. The mutant displays motility in vitro, so presumably this is because it lacks proper/robust interaction with free tubulin dimers. Although this could not be tested directly.

Overall this is a comprehensive body of work dissecting the MT polymerization and stabilizing activity of Kip2 in relation to Bik1 both in vivo and in vitro. It is well written, flows nicely, and data are high quality. The various mutants used are quite informative and the data support a model of Bik1 recruiting Kip2 to the SPB, Kip2 transporting Bik1 to the +end, then associating with the Bik1 pool at the plus end while using its motor domains to promote elongation of the +end. This is a significant conceptual advance regarding Kip2/Bik1 function. The data fall short of sufficiently supporting certain mechanistic aspects of the model.

Concerns:

- 1) A major mechanistic component of the model, that Kip2 motor domain binds curved tubulin to promote MT polymerization depends on the P1 mutant not binding to free tubulin. This should be tested at least in the SEC-MALS experiment as in Fig 5A. Moreover, the explanation on page 12 of why this interaction was not tested is unclear. Since this could not be done due to precipitation of the Kip2-MD-P1 mutant in tubulin buffer, and presumably a similar tubulin buffer was used for the TIRF and motility assays. Yet the SEC-MALS was in a Tris-based buffer.
- 2) The issue of precipitation also raises questions about the TIRF based experiments which support the direct involvement of P1 in MT polymerization. Equal molar concentrations of WT and P1 mutant were compared, with WT displaying more activity. P1 precipitation will result in lower apparent activity. How was the precipitation of the P1 mutant controlled for in the effective concentration? Is it known to what extent P1 was soluble under these conditions?
- 3) Why does tubulin elute from the SEC column at 13.75 ml in Fig 2 and at 9.4 ml in Fig 5? Furthermore, why is Kip2-MD alone not analyzed in Fig 5? It is not possible to judge the strength of this interaction.
- 4) The authors use an automated system to measure MT length at ~1s intervals based on the distance between the centroid of Spc72 and Bik1 signals. It is unclear whether growth and depolymerization rates were calculated based on single timepoint changes in calculated length. More importantly, the criteria for how rescues and catastrophes were calculated based on these data is unclear. If they were calculated based on changes from positive to negative length changes within single timepoints, for instance, how can the authors control for potential changes in the shape and amount of Bik1 at the +end? Do the increased catastrophe events in kip2 delta cells lead to short-lived depolymerization events or more commonly to complete MT depolymerization?
- 5) The cells in Fig S1B should be visualize using two colors as in Fig S1A. Particularly with the short spindle and aMT it is uncertain whether the two main spots represent 1 SPB and 1 +end or 2 SPBs. What is indicated as the second SPB is poorly visualized and also changing position. If this is due to 3D movements it clearly illustrates the limitations of single color imaging for multiple cellular structures. This result is critical for the interpretation of the Kip3-Kip2 chimera results in Fig 1D and the resulting conclusion that bringing the Kip2-NMD to the +end is not sufficient to restore MT growth without Kip2 motor domain.
- 6) The data in Fig 4 A and C only show 1-2 cells per condition to back up conclusions in the text and needed to support subsequent findings. These results need to be quantified from a reasonable number of cells over multiple experiments. The images shown in Fig. 4C are also less than convincing that full length Kip2 is located at the MT +end in these cells.
- 7) The authors use labels at the 2 ends of the MT to infer length. Visualizing the MT may be slower but has the advantage of detecting MT bending, which can occur with longer MTs (e.g. in DeltaN cells). The authors should exclude the influence of MT bending from the conclusions and/or determine the frequency by also visualizing MTs in some cells.

Minor issues:

8) Why was the recruitment of Kip2-N-MD to SPB (or +end) by Bik1 not tested?

9) The authors should include some representative graphs of the MT length and lifetime measured with the method used in Fig 1.

10) In Fig 1 it is unclear if the graphs report maximum MT length of each MT, or the lengths of MTs accumulated from each time frame. The text indicates max length only but the figures and legend seem otherwise. Also, it should be clearly stated whether, if max length is graphed, how the authors eliminated multiple maximums following rescue events.

11) Fig. 3G. The graphs are somewhat busy, and so it should be specifically noted in the text as for the statistical significance of Kip2-GFP in *bik1*Δ cells, the results for the ΔT cells, while similar effects, changes in neither parameter are statistically significant. The present wording could be somewhat misleading.

12) The data/model presented in this manuscript would be strengthened by context with the recently demonstrated remote control mechanism regulating Kip2 loading and transport on MTs (Chen et al., 2019, eLife). In this manuscript it was shown that phosphorylation of Kip2 N region restricted Kip2 loading to SPBs. Is this mechanism independent of those shown in the current manuscript, upstream, or perhaps synergistic?

Reviewer #1 (Comments to the Authors (Required)):

The microtubule plus-end directed molecular motor Kip2, in combination with the CLIP-170 homolog Bik1, has been shown to be important for cytoplasmic microtubule dynamics in budding yeast. The molecular mechanisms by which Kip2 promotes tubulin polymerization in cells and the contribution of Bik1 during this process, however, are not fully understood. In the current study, the authors use a combination of live cell imaging with high temporal resolution, in vitro biochemical and biophysical experiments and mathematical modelling to interrogate new aspects of the Kip2-Bik1 system. They show that the Kip2-NMD (non-motor domain) is sufficient for this interaction and that it depends on the c-terminal tail of the Kip2-NMD. Plus-end polymerization activity in vivo requires the motor domain of Kip2 and the authors detect binding to tubulin dimers in vitro. Structural homology models allow the authors to propose three positively charged patches on the motor domain to be selectively involved in microtubule lattice binding, versus binding to tubulin dimers. Interestingly, mutation of patch 1 (P1), allows motility in vitro, plus-end localization in vivo, yet seems to lack polymerization activity. The authors propose a model in which Kip2 promotes the incorporation by binding to free tubulin dimers, once it has reached the plus-end. Bik1 increases the retention of Kip2 at the plus-end to support this effect.

Overall this is an interesting study with a nice combination of in vivo and in vitro analysis. While there has been a fair amount of work on Kip2 already, there are a number of interesting novel aspects in this study.

We thank the reviewer for carefully reading our manuscript. We are very happy that the reviewer finds it interesting and appreciates the novel aspects revealed by our study.

Major points:

1. A novel aspect of the current study is the identification of the distinct patches P1, P2, P3 on the Kip1 MD involved in the binding of microtubules versus tubulin dimers. I missed the demonstration that these mutants indeed affect tubulin dimer binding in vitro (like in Figure 5A). This seems to me a crucial point in the current study, especially regarding the effect of the P1 mutation.

We have now succeeded in purifying the recombinant proteins MBP-Kip2-MD-mCherry and the corresponding mutant MBP-Kip2-MD-P1-mCherry. Although the motor domain (MD) alone was difficult to work with, particularly when mutated, fusing it N-terminally to MBP and C-terminally to mCherry stabilized it and allowed us to work with it. SEC-MALS analysis of these proteins in the presence of unpolymerized tubulin are now shown in Figure 5CD. Beyond confirming that Kip2-MD binds unpolymerized tubulin, we now show that the P1 patch is required for this interaction.

2. The use of an in vitro reconstitution assay for tubulin polymerization to demonstrate an effect of the Kip2 P1 mutation (Figure 6) is appreciated. Along these lines it is a bit of a pity that the first part of the study, which mapped crucial elements of the Kip2-Bik1 interaction, is not connected to these in vitro experiments in a similar manner. The authors cite that Bik1 is not required for Kip2 mediated polymerization in vitro, but there might be a quantitative effect of Bik1 in the reconstituted system? It seems that more features of the model proposed in Figure 7 could be tested in this setup.

We agree with the reviewer that it would be good to demonstrate the effect of Bik1 on Kip2-mediated microtubule polymerization in vitro quantitatively. To address this point, we purified full length Kip2 fused to GFP and full length Bik1 fused to mCherry. In TIRF experiments with Bik1

and Kip2, we found that Bik1 binds microtubules poorly - less than half of the microtubules have bound Bik1. This is consistent with previous reports and could be explained in several manners. First, Bik1 binds C-terminal phenylalanine residues as in yeast tubulin but very poorly the C-terminal tyrosine of porcine tubulin used in our assay (see M. M. Stangier et al., 2018). This alone could already explain the difficulties of Bik1 binding microtubules in our in vitro assay. However, additional challenges affect this experiment further. CLIP proteins require EBs to track microtubule ends (K. A. Blake-Hodek, et al., 2010). The addition of Bim1 to the experiment and analysis of its own role would therefore be needed, which goes beyond the scope of this study. In any case, the addition of Bik1 in the tubulin polymerization assay did not affect microtubule dynamics, not surprisingly given these limitations. Given the inconclusive nature of these experiments, we have not added them to the manuscript.

Minor points:

Figure 2 B,D,F: It would be good to show the corresponding SDS PAGE, this could be done in the supplement.

In the SEC-MALS experiments shown in Figure 2 B, D, and E, the SEC analysis is accompanied by the continuous determination of molecular weights of the molecular species eluting from the SEC column. It is thus unusual to show SDS-PAGEs of fractions collected along a SEC-MALS experiment on top and indeed we did not collect any corresponding fractions. To accommodate this reviewer's request, we would need to repurify all proteins and reanalyze all samples again by SEC-MALS, which in our opinion will not change our conclusions. However, we now show in the new Figure S1E presenting SDS-PAGE analyses of the proteins used in our biochemical studies (including the ones used in our new TIRF experiments).

Figure 2E: it would be nice to show raw ITC data as heat curves, this could be done in the supplement.

Since the ITC data did not add substantial further insights to our main conclusion that "Bik1-CC interacts directly with Kip2-NMD and this stable interaction depends on Kip2's C-terminal tail" (page 7, line 4-5 of the revised manuscript), we have simplified the text and removed these data from the manuscript. The raw data are therefore no-longer relevant.

Figure 3: I found the order of the B and C to be confusing. Better to show the wildtype first?

We agree with the reviewer and are now showing wild type first.

Figure 5 A: I missed data for Kip2 MD alone and a corresponding analysis by SDS-PAGE of the SEC fractions.

We no longer have access to Kip2-MD proteins, which were very unstable, but have been able to produce MBP-Kip2-MD-mCherry. As mentioned above, we now show the data obtained with it in the new Figure 5C and Figure S1E.

I found the usage of the terms Kip2 NMD and Kip2 tail sometimes confusing. Sometimes NMD and Tail seemed synonymous for the non-motor part, sometimes "tail" only meant the extreme carboxyterminus. The authors should check again for consistency throughout the manuscript.

We now clearly define and only refer to the extreme C-terminus as the Kip2 tail.

Page 5: Heading of the section: one should qualify the statement with "astral" microtubule growth in vivo

We inserted 'astral' in the heading.

Reviewer #2 (Comments to the Authors (Required)):

In this manuscript, Chen et al. continue their investigation of the budding yeast kinesin polymerase Kip2, focusing on elucidating the molecular mechanisms underlying its microtubule polymerase activity. They employ a range of methods, from live-cell imaging of budding yeast, through biochemical and structural approaches to probe protein interactions, to TIRF-based reconstitution assays with purified protein components. The authors report that the c-terminal tail of Kip2 mediates interaction with its binding protein Bik1 (yeast Clip-170 ortholog), and that this tail domain is important for the Kip2's polymerase activity in vivo. They postulate that Bik1-Kip2 interaction contributes to Kip2's microtubule plus-end tracking and retention at plus ends in vivo. On the other hand, the authors report that Kip2's motor domain binds soluble tubulin, and that the residues involved in tubulin interaction (K294 & R296) are necessary for Kip2's polymerase activity in vitro. Taken together, the authors propose that Kip2-Bik1 interaction facilitates prolonged residency of Kip2 at the microtubule plus end, where motor domains of Kip2 facilitate microtubule polymerization through a direct interaction with soluble tubulin dimers.

The study is of high quality, and the results are presented in a clear and logical fashion. Several interesting questions and controversies remain. For example, in their previous work (Chen et al. 2019), the authors reported that Kip2 targeting to spindle pole bodies (SPB) depends on Bub2 and Bfa1, while Bik1 primarily targets microtubule plus ends. In this work, however, the authors conclude that Bik1 promotes Kip2 recruitment to SPBs (see Discussion), although it is not completely clear how that would work. Additionally, the conclusion that Bik1 increases Kip2's residency time at microtubule plus ends is not directly demonstrated, and the corresponding conclusions should thus be toned down. Finally, the authors are in a position to directly disprove a competing tubulin-shuttle model (suggested by Hibbel et al. 2015), which would provide significant support to the model proposed in this manuscript.

We thank the reviewer very much for carefully reading and commenting on our manuscript. We are very pleased that the reviewer praises the quality of our study. We address below the specific concerns raised.

We have indeed previously demonstrated the function of Kip2 as a messenger deployed by yeast spindle pole bodies (SPBs) to control astral microtubule length and dynein distribution (Chen et al., 2019). In that study, we reported that Kip2 recruitment at the SPBs depends on Bub2 and Bfa1, but we did not investigate the contributions of Bik1. Therefore, that study did not exclude the possibility that Bik1 participates in any of the processes of Bik1 recruitment at SPBs and stabilization at microtubule tips. Indeed, Bik1 and Kip2 are both found at SPBs in addition to the plus-end of astral microtubules (Carvalho P. et al., 2004). It has been proposed that Kip2 meets Bik1 at the SPB and then transports it to microtubule plus-ends. But how Bik1 affects Kip2 localization and functions has been unclear, and this is one of the issues we address in this work. See our specific responses to the distinct points of the reviewer below.

Specific comments:

Figure 1:

- It would be very insightful to show an example image/kymograph from which microtubule dynamics parameters (lifetime, growth and shrinkage rates) are extracted

We thank the reviewer for this suggestion. We have inserted two new panels in Figure 1. Figure 1D shows an example of a preanaphase astral microtubule accompanied by kymographs and tracked trajectories. Figure 1E shows the corresponding plot of 3D aMT length as a function of time. We highlight how the maximum microtubule length, lifetime, speeds of growth, and shrinkage were extracted. We also added some examples of 3D astral microtubule length plotted as a function of time extracted from control and *kip2Δ* mutant cells in Figure S1AB.

- Representation of microtubule growth and shrinkage rates on a log plot (Figure 1G) is rather unintuitive - the authors should instead plot microtubule growth rates and microtubule shrinkage rates separately, both on a linear plot.

As suggested, we plotted the speeds of the microtubule growth and shrinkage on a linear scale.

- Numbers of measurements should be added to the caption for all data (e.g. panels 1F and 1G)

We updated the numbers of measurements on the panels.

Figure 2:

- Panels are incorrectly labeled.

Thank you for spotting this. The labels are now corrected.

Figure 3:

- This figure is the most confusing, while at the same time essential for the conclusions on how Kip2-Bik1 interaction encodes Kip2 localization in vivo. Since the intention is to directly compare wt vs *delT* Kip2 constructs in the presence and absence of Bik1, it would be more useful in 3B-D to plot intensity distributions for different mutants against each other on the same graph (e.g. just take 2.5 μm long bin and plot the three distributions together)

Thank you for the suggestion, which we considered, but ultimately did not implement in the main text for the following reasons:

- First, the distributions of Kip2 along aMTs differ not only between wt and mutants but also between aMTs of different lengths, such that picking only one length bin would neglect other important aspects of the data (please see also response below).
- Second, we plotted these three graphs with consistent x- and y-scales to allow readers to compare across graphs. We now include an explicit statement when first referencing Fig. 3B-D.
- Finally, we now present the intensity distributions per length bin, as suggested by the reviewer, in Supplementary Figure 3A to F.

- The significance of slopes and intercepts in 3B-D is not discussed.

We now discuss the significance in the context of in vivo vs in vitro residence times at aMT plus-ends (see below).

- While the authors have previously employed and extensively discussed their model (Chen et al. 2019), the significance of the modeling results in 3F-H is not clear and sufficiently elaborated. Modeling results should be overlaid with experimental data for direct comparison, as previously done in Chen et al. 2019. It is unclear to what extent the model predicts a unique set of effects on the modeled parameters. For example, could e.g. obtained differences in in-rate be compensated by differences in on-rate and/or concentration at SPBs? Is the 30% reduction in SPB body localization (established using ATP-deficient mutant) explicitly modeled?

As suggested, we now provide an overlay of experimental data and simulation results (Supplementary Figure 3G to I). Comparisons of simulation and experimental data obtained for the *WT* and *bik1Δ* mutant cells indicate near-perfect matches, but peak densities for Kip2- Δ T-3xsfGFP cells show deviations. Overall we conclude that the model adequately captures the data (see revised main text), and discuss the deviation in relation to the reviewer's other points on slopes/intercepts (above) and (Hibbel et al., 2015) (below).

Regarding unique sets of parameters, we have now added the following clarification: 'note that we used a Bayesian approach to estimate parameter values and their uncertainties; these uncertainties may be caused by measurements noise or by limited identifiability of individual parameters.' The justification for this statement is that, if the model were not identifiable, the kernel density estimates would indicate this. For example, k_{on} and k_{in} are harder to identify because the on / in rates depend on these parameters as well as the concentration of free Kip2 multiplicatively. As in our previous work (Chen et al, 2019), the key to identifiability is to fix Kip2's movement speed, which according to Fig. S2B is unchanged between wt and mutants at approximately 6.3 $\mu\text{m}/\text{min}$ (consistent with (Chen et al, 2019)).

For the 30% reduction in SPB localization of Kip2, with the simplified model (the SPB is not modeled as an explicit compartment) and the available data (which cannot disentangle local Kip2 concentration at the SPB and transition rate constant to the microtubule), we do not see how to capture this effect in a principled manner. The current model captures only the total in-rate and it is not clear how mutations influence the transition from the SPB. We now state this limitation explicitly.

- Results presented in 3G should be plotted separately on a linear (not log) scale for clarity.

We now provide a better explanation for this plot in log-scale and apologize for the omission. We added the statement 'The modelling results first showed that in wild type and both mutants, Kip2's on-rate constants were orders of magnitude lower than its in-rate constants, previously observed (37). This implies that the primary recruitment of Kip2 from the microtubule minus-end (i.e., the SPB) is not fundamentally affected upon removing Bik1 or preventing Kip2 to interact with it. This interpretation is essential for the following discussion of differences between wt and mutants, and it requires a comparison of k_{in} and k_{on} estimates that is impossible in linear scale in one plot (together with differentiation between wt and mutants).

- Importantly, while 3H shows that modeling estimates a 2-fold increase in Kip2 out rate in the absence of Bik1 interaction, which corresponds to end-residency being reduced from 0.3 s to ~0.15 s, previous in vitro results (Hibbel et al. 2015) demonstrated that Kip2 residency times on plus ends on its own in vitro are two orders of magnitude higher (~30 s). Given such a large discrepancy between in vitro and in vivo results, it is hard to appreciate the potential effect that just a 2-fold modulation of the out rate might have.

To address the discrepancy of residence times with in vitro data, we first simulated the model with the k_{out} value from (Hibbel et al., 2015), and all other parameter values unchanged. This resulted in predictions of ‘traffic jams’ at aMT plus-ends as shown in the simulated kymograph in the new Supplementary Figure 3J, which are clearly inconsistent with our in vivo data for Kip2.

For the related kinesin-8 Kip3, it is known that MT plus-end residence time decreases exponentially with (total) Kip3 concentration (1) and with mechanical force applied to these motors (2). Importantly, the in vitro experiment to quantify residence time in (Hibbel et al., 2015) used <1nM total Kip2, one to two orders of magnitude lower than our in vivo estimates (Fig. 3F). If Kip2 had a similar concentration and force dependence as Kip3, this would explain the difference between in vivo and in vitro residence times. It would also be consistent with (i) Decreasing peak densities (indicated by slopes in Fig. 3B-D) for (long) microtubules with high Kip2 density (indicated by intercepts in Fig. 3B-D), and (ii) mismatches between model and experimental data, particularly for long aMTs in Kip2- Δ T-3xsfGFP cells where these uncaptured concentration effects would be most pronounced. We now provide this hypothesis and reasoning in the main text.

Figure 4:

- The data in this figure (especially panel C) would be well supported by accompanying intensity linescans, allowing to directly compare investigated protein distributions/co-localizations.

We have generated and added line scan plots for panel C, now presented in panel D.

- It is a little bit confusing that Kip2-NMD is targeted to plus ends via Bik1 interaction, but the full-length ATP-ase deficient Kip2 does not localize to plus ends in the presence of Bik1. Shouldn't this mutant still be able to bind Bik1 through its tail domain (have the authors checked this)? How is this observation explained?

Indeed, the ATPase deficient Kip2 variant Kip2-G374A does not localize to plus-ends in the presence of Bik1. As shown in Figure 3C in Chen et al., 2019, in *KIP2-G374A-3xsfGFP/KIP2-mCherry* heterozygous diploid cells, Kip2-mCherry is enriched at the plus-end, and Kip2-G374A-3xsfGFP is absent from microtubule plus-ends. One of the main conclusions from that work is that Kip2 recruitment to microtubules in yeast cells is highly restricted to the SPBs (Chen. et al., 2019). Together, our previous and current data suggest that full length Kip2 cannot bind Bik1 at the microtubule plus-end (or the shaft for that matter) without first passing by the SPB. Now, the fact that Kip2-NMD can bind Bik1 at and localize to the plus-end of microtubules on its own comes indeed as a surprise. We suggest that some of the sequences present in the full length Kip2 and absent in the NMD construct restrict Kip2 from binding Bik1 at microtubule ends as long as it has not passed first by the minus end. For example, the disordered N-terminal domain of Kip2 (about 100 amino acids) and its phospho-regulation contribute to focusing the proteins recruitment to microtubules minus-ends (Chen et al., 2019). This domain is absent in the NMD construct. This might explain why the NMD and full length Kip2 show different pre-requirements for them to bind Bik1 at microtubule plus-ends.

Figure 7:

- The authors propose a model in which Kip2-soluble tubulin dimer interaction is only possible if a motor head is not engaged with the microtubule lattice, restricting it to microtubule plus ends. To further eliminate the possibility that Kip2 is directly shuttling tubulin (as proposed in Hibbel et al. 2015), the authors should perform TIRF reconstitution experiments (like in Figure 6) at much lower Kip2-GFP and labeled tubulin concentrations (i.e. 100 pM-1 nM Kip2 and sub-micromolar

tubulin range) which would directly reveal whether Kip2-tubulin interactions can occur along the microtubule lattice, and/or only at the ends).

This is an interesting point. However, we are not ourselves convinced that our model and the shuttling model would be necessary mutually exclusive. The fact that the motor domain of Kip2 is able to directly bind and polymerize tubulin dimers at microtubule tips does not a priori exclude another part of the molecule from binding and transporting such dimers towards the microtubule tip. In any case, we have tried but not been able to visualize single molecules of tubulin dimers in TIRF experiments so far. Therefore, we cannot exclude that Kip2 would also transport tubulin dimers.

- The evidence that Bik1 prolongs the residence time of Kip2 at ends is weak, as no direct measurements of residence times have been performed. This, of course, could be done using in vitro TIRF approach with purified Bik1. This may well be outside the scope of the manuscript, but in that case the authors should restrain from strong statements that this is indeed what is going on in cells (see Discussion).

We attempted to directly demonstrate that Bik1 increases Kip2's residency time at microtubule plus ends in vitro. We have purified full length Kip2 fused to GFP from SF9 insect cells and full length Bik1 fused to mCherry from bacteria. However, in TIRF experiments we found that Bik1 binds to microtubules poorly. This is consistent with previous reports (K. A. Blake-Hodek, et al., 2010 and M. M. Stangier et al., 2018). This is a major technical obstacle that prevents us from performing this experiment (see also response to point 2 of reviewer 1). Therefore, we have not added these experiments to the paper.

This being said, we do not think that the evidence that Bik1 prolongs the residence time of Kip2 at microtubule tips is that weak. Just looking at the images obtained in vivo clearly shows that the strong localization of Kip2 focused to the plus-end of astral microtubules in vivo disappears when Kip2 no-longer interacts with Bik1 (*bik1* Δ and *Kip2- ΔT* mutant cells). Thus, Kip2 does stick for some time to the plus-end of microtubules and this requires Bik1 function. The simulations clearly support this notion and put numbers on it. Even if the effect might not seem huge, the point is strong: although the amount of Kip2 at microtubule tips is higher than in wild type due to an increased flow of Kip2 molecules towards the plus-end, in the *bik1* Δ mutant cells Kip2 leaves virtually as it arrives to the plus-end and this is sufficient for causing a consequent defect in microtubule polymerization. Thus, there is little doubt that Bik1 retains Kip2 at ends and that this is essential for Kip2 to function as a polymerase in vivo.

Reviewer #3 (Comments to the Authors (Required)):

The manuscript from Chen et al. addresses the important question of how kinesin proteins regulate microtubule (MT) dynamics overall, and specifically how the kinesin Kip2 promotes MT polymerization and stabilization. This activity is unclear because Kip2 requires its partner Bik1 to promote MT growth/stability in vivo, yet is capable of this alone in vitro. Bik1 itself is required for MT growth/stability in vivo. This manuscript investigates the mechanism of how Kip2 promotes MT growth and stability. The authors examine MT dynamics in cells and show that Kip2 likely promotes both MT polymerization and inhibits catastrophes. The various Kip2 truncations used are informative for the molecular regions required for MT growth, recruitment to SPBs, interaction with tubulin and with Bik1. The authors also test binding interactions between these regions and tubulin, MTs, and parts of Bik1 (CC domain). The authors find no evidence of binding of Kip2-NMT domain to tubulin or MTs using the construct from bacteria and conditions used. Interestingly, they do find that the CC domain of Bik1 does not interact with the CC of

Kip2, but rather mainly via the tail region. Using a combination of in vivo imaging and mathematical modeling the authors present data that Bik1 likely recruits Kip2 to SBPs for transport along MTs, and prevents Kip2 from loading along the MT lattice. Bik1 also seems required for Kip2 remaining associated with MT +ends. Data also shows that the NMD of Kip2 recruits Kip2 to the SBP via Bik1, and that this effect may not occur directly at the +end, hence, Kip2 and Bik1 interactions originate the SPB. The authors identify three regions of positive charge on the MT-binding surface and mutate them, showing that P2 and P3 mutants fail to localize properly, whereas P1 mutants can. Although the P1 mutant localizes to +ends, the MTs are short as in a kip2 null. Thus, the P1 patch is needed to promote MT polymerization in vivo. The authors then test the function of Kip2-MD-P1 compared to WT Kip2-MD on MT polymerization and stability in vitro, with data suggesting that the P1 mutant is inefficient at promoting MT growth and stability. The mutant displays motility in vitro, so presumably this is because it lacks proper/robust interaction with free tubulin dimers. Although this could not be tested directly.

Overall this is a comprehensive body of work dissecting the MT polymerization and stabilizing activity of Kip2 in relation to Bik1 both in vivo and in vitro. It is well written, flows nicely, and data are high quality. The various mutants used are quite informative and the data support a model of Bik1 recruiting Kip2 to the SPB, Kip2 transporting Bik1 to the +end, then associating with the Bik1 pool at the plus end while using its motor domains to promote elongation of the +end. This is a significant conceptual advance regarding Kip2/Bik1 function. The data fall short of sufficiently supporting certain mechanistic aspects of the model.

We thank the reviewer for these very positive words. We agree with the points raised by the reviewer, which we addressed in full.

Concerns:

1) A major mechanistic component of the model, that Kip2 motor domain binds curved tubulin to promote MT polymerization depends on the P1 mutant not binding to free tubulin. This should be tested at least in the SEC-MALS experiment as in Fig 5A. Moreover, the explanation on page 12 of why this interaction was not tested is unclear. Since this could not be done due to precipitation of the Kip2-MD-P1 mutant in tubulin buffer, and presumably a similar tubulin buffer was used for the TIRF and motility assays. Yet the SEC-MALS was in a Tris-based buffer.

We now made new constructs, adding an MBP tag at the N-terminus and a mCherry tag at the C-terminus. These changes substantially increased the solubility and stability of the proteins, allowing us to successfully purify the recombinant proteins MBP-Kip2-MD-mCherry and MBP-Kip2-MD-P1-mCherry. We performed SEC-MALS analysis of these proteins in the presence of free tubulin. The results are shown in Figure 5CD. We demonstrate that Kip2 MD indeed binds free tubulin and that the P1 mutation impairs Kip2-tubulin complex formation.

2) The issue of precipitation also raises questions about the TIRF based experiments which support the direct involvement of P1 in MT polymerization. Equal molar concentrations of WT and P1 mutant were compared, with WT displaying more activity. P1 precipitation will result in lower apparent activity. How was the precipitation of the P1 mutant controlled for in the effective concentration? Is it known to what extent P1 was soluble under these conditions?

The Kip2-MD proteins used in the SEC-MALS experiment shown in Figure 5A were not tagged and comprise only the motor domain alone. The TIRF experiments shown in Figure 6A-E were performed with MBP-Kip2 [1-560]-RFP. Not only these versions of the protein are fused to MBP

and RFP, they are also longer than Kip2-MD [100-503]. This was refereed in pg.11 , now pg12): “At 2 nM concentration, purified wild type MBP-Kip2(1-560) fused to red fluorescent protein (RFP; Fig. 6AC; Table S2 and Fig. S1E, denoted Kip2-WT) strongly increased ...”. Under our experimental conditions, we did not observe protein precipitation with any of the recombinant proteins reported in the manuscript.

3) Why does tubulin elute from the SEC column at 13.75 ml in Fig 2 and at 9.4 ml in Fig 5? Furthermore, why is Kip2-MD alone not analyzed in Fig 5? It is not possible to judge the strength of this interaction.

We used a Superdex200 SEC column for all but the experiment shown in Figure 5A, which was performed with a Superdex75 SEC column. The method section has been completed accordingly. For what concerns MBP-Kip2-MD-mCherry, we now show the analysis of the protein alone in Figure 5C.

4) The authors use an automated system to measure MT length at ~1s intervals based on the distance between the centroid of Spc72 and Bik1 signals. It is unclear whether growth and depolymerization rates were calculated based on single timepoint changes in calculated length. More importantly, the criteria for how rescues and catastrophes were calculated based on these data is unclear. If they were calculated based on changes from positive to negative length changes within single timepoints, for instance, how can the authors control for potential changes in the shape and amount of Bik1 at the +end? Do the increased catastrophe events in kip2 delta cells lead to short-lived depolymerization events or more commonly to complete MT depolymerization?

We use a semi-automated system to record and document the dynamics of microtubules in living cells. The catastrophe and rescue events were annotated manually based on the global microtubule length profile rather than length changes between two time points (1.07 to 2.14 s). We now present representative examples of 3D astral microtubule length plotted as a function of time extracted from control and kip2del cells in Figure S1AB. We also state that the catastrophe and rescue events were manually annotated. Increased catastrophe events in kip2Δ cells more commonly lead to complete microtubule depolymerization.

We also inserted two new panels in Figure 1. Figure 1D shows an example of a pre-anaphase astral microtubule accompanied by kymographs and tracked trajectories. Figure 1E shows the corresponding plot of 3D aMT length as a function of time. We highlight how the maximum microtubule length, lifetime, speeds of growth, and shrinkage were extracted.

5) The cells in Fig S1B should be visualized using two colors as in Fig S1A. Particularly with the short spindle and aMT it is uncertain whether the two main spots represent 1 SPB and 1 +end or 2 SPBs. What is indicated as the second SPB is poorly visualized and also changing position. If this is due to 3D movements it clearly illustrates the limitations of single color imaging for multiple cellular structures. This result is critical for the interpretation of the Kip3-Kip2 chimera results in Fig 1D and the resulting conclusion that bringing the Kip2-NMD to the +end is not sufficient to restore MT growth without Kip2 motor domain.

All our time-lapses were recorded in single color to achieve high temporal resolution. Very few haploid Kip3-Kip2 chimera cells made visible aMTs. For the cell shown in Fig S1B, now Fig S1D, the microtubule plus-end started above the bud-neck at the first time point. Shrinkage of the microtubule then brought the plus-end across the bud neck down to the mother cell. In contrast, the SPBs were less mobile and never crossed the bud neck into the bud. To strengthen our

evidence, we now include a representative movie (movie S2) of the cell shown in Fig S1D. In the movie, the microtubule plus-end and the spindle poles can be differentiated with excellent confidence.

6) The data in Fig 4 A and C only show 1-2 cells per condition to back up conclusions in the text and needed to support subsequent findings. These results need to be quantified from a reasonable number of cells over multiple experiments. The images shown in Fig. 4C are also less than convincing that full length Kip2 is located at the MT +end in these cells.

These results illustrated in figure 4A are now quantified for substantial cohorts of cells and reported in the new Figure 4B.

Images for Kip2-G374A-3xsfGFP/Kip2-mCherry diploid cells in the previous Figure 4C (now Figure 4D) were challenging to acquire since mCherry is not as bright as 3xsfGFP. We increased the exposure time for the mCherry channel from 30 ms to 50 ms. We now provide the line scan analysis of the microtubules to demonstrate the relative localization of Kip2-G374A-3xsfGFP and Kip2-mCherry.

7) The authors use labels at the 2 ends of the MT to infer length. Visualizing the MT may be slower but has the advantage of detecting MT bending, which can occur with longer MTs (e.g. in DeltaN cells). The authors should exclude the influence of MT bending from the conclusions and/or determine the frequency by also visualizing MTs in some cells.

This is indeed an important point. We have also visualized these microtubules using tubulin labels. Among all mutants reported in this work, aMT bending was only observed in Kip2-ΔN pre-anaphase cells. We excluded those cells from our analysis and clarified this point in the methods section. Since including those cells will only further increase the length of aMTs in Kip2-ΔN pre-anaphase cells, it would not change our conclusions.

Minor issues:

8) Why was the recruitment of Kip2-N-MD to SPB (or +end) by Bik1 not tested?

We did not test this mutant in living cells as Kip2-N-MD would not dimerize. We did express Kip2-N-3xsfGFP from the endogenous *KIP2* locus, the resulting proteins were diffusive and did not localize to any specific structure.

9) The authors should include some representative graphs of the MT length and lifetime measured with the method used in Fig 1.

We followed this excellent idea and inserted panel DE to Figure 1 to illustrate how the maximum aMT length, lifetime, and speeds of growth and shrinkage were extracted. We further present more examples of 3D aMT length as a function of time in Figure S1.

10) In Fig 1 it is unclear if the graphs report maximum MT length of each MT, or the lengths of MTs accumulated from each time frame. The text indicates max length only but the figures and legend seem otherwise. Also, it should be clearly stated whether, if max length is graphed, how the authors eliminated multiple maximums following rescue events.

Indeed, we report maximum aMT length in our tables and graphs. For each aMT, we generate only one value of maximum length, regardless of the number of rescue events. We have now clarified this in our captions and methods.

11) *Fig. 3G. The graphs are somewhat busy, and so it should be specifically noted in the text as for the statistical significance of Kip2-GFP in bik1delta cells, the results for the deltaT cells, while similar effects, changes in neither parameter are statistically significant. The present wording could be somewhat misleading.*

Thank you for spotting this imprecision – we have revised the statement.

12) *The data/model presented in this manuscript would be strengthened by context with the recently demonstrated remote control mechanism regulating Kip2 loading and transport on MTs (Chen et al., 2019, eLife). In this manuscript it was shown that phosphorylation of Kip2 N region restricted Kip2 loading to SPBs. Is this mechanism independent of those shown in the current manuscript, upstream, or perhaps synergistic?*

In (Chen et al., 2019, eLife), we reported that phosphorylation of Kip2's N-terminus prevents Kip2 from landing along microtubule lattices by ablating phosphorylation with a S63A point mutation (Kip2-S63A). In the current study, we removed the N-terminus to generate the mutant Kip2-ΔN. Both mutants presented longer and longer-lived astral microtubules, which suggests that Kip2's N-terminus negatively regulates Kip2's microtubule polymerization activity. Since the S63A mutant shows a similar phenotype, our data suggest phosphorylation represses the inhibitory role of the N-terminus. Interestingly, the inhibitory activity of Kip2's N-terminus is no longer observed in the absence of Kip2's C-terminal tail (Figure 1CF). Fusing the N-terminus upstream of the NMD fragment dampened the ability of Kip2-NMD to target microtubule plus-ends and SPBs (unpublished result). This effect was reduced when the serine S63 was mutated to alanine. We can conclude that Kip2's N-terminus is a negative regulator of the microtubule polymerase activity of Kip2 and of Kip2 accumulation at microtubule tips. What the mechanism of this inhibition is and how it relates to the phosphorylation-based restriction in Kip2 loading onto microtubule shaft are open questions that will require extensive work and extend beyond the scope of the present study.

February 21, 2023

RE: JCB Manuscript #202110126R

Prof. Yves Barral
ETH Zurich
Biology
Institute of Biochemistry ETH Zurich Schafmattstrasse 18
Otto-Stern-Weg 3
ZUERICH 8093
Switzerland

Dear Prof. Barral,

Thank you for submitting your revised manuscript entitled "The motor domain of the kinesin Kip2 promotes microtubule polymerization at microtubule tips." We would be happy to publish your paper in JCB pending final revisions necessary to meet our formatting guidelines (see details below) as well as the text changes requested by Reviewer #3 to more clearly differentiate between conclusions based on data and those based on modeling simulations.

A. MANUSCRIPT ORGANIZATION AND FORMATTING:

1) Text limits: Character count for Articles is < 40,000, not including spaces. Count includes title page, abstract, introduction, results, discussion, and acknowledgments. Count does not include materials and methods, figure legends, references, tables, or supplemental legends.

2) Figure formatting: Articles may have up to 10 main text figures. Scale bars must be present on all microscopy images, including inset magnifications. Molecular weight or nucleic acid size markers must be included on all gel electrophoresis. Please add MW markers to Figure S2D.

Also, please avoid pairing red and green for images and graphs to ensure legibility for color-blind readers. If red and green are paired for images, please ensure that the particular red and green hues used in micrographs are distinctive with any of the colorblind types. If not, please modify colors accordingly or provide separate images of the individual channels.

3) Statistical analysis: Error bars on graphic representations of numerical data must be clearly described in the figure legend. The number of independent data points (n) represented in a graph must be indicated in the legend. Please, indicate whether 'n' refers to technical or biological replicates (i.e. number of analyzed cells, samples or animals, number of independent experiments). If independent experiments with multiple biological replicates have been performed, we recommend using distribution-reproducibility SuperPlots (please see Lord et al., JCB 2020) to better display the distribution of the entire dataset, and report statistics (such as means, error bars, and P values) that address the reproducibility of the findings.

Statistical methods should be explained in full in the materials and methods. For figures presenting pooled data the statistical measure should be defined in the figure legends. Please also be sure to indicate the statistical tests used in each of your experiments (both in the figure legend itself and in a separate methods section) as well as the parameters of the test (for example, if you ran a t-test, please indicate if it was one- or two-sided, etc.). Also, if you used parametric tests, please indicate if the data distribution was tested for normality (and if so, how). If not, you must state something to the effect that "Data distribution was assumed to be normal but this was not formally tested."

4) Materials and methods: Should be comprehensive and not simply reference a previous publication for details on how an experiment was performed. Please provide full descriptions (at least in brief) in the text for readers who may not have access to referenced manuscripts. The text should not refer to methods "...as previously described."

5) For all cell lines, vectors, constructs/cDNAs, etc. - all genetic material: please include database / vendor ID (e.g., Addgene, ATCC, etc.) or if unavailable, please briefly describe their basic genetic features, even if described in other published work or gifted to you by other investigators (and provide references where appropriate). Please be sure to provide the sequences for all of your oligos: primers, si/shRNA, RNAi, gRNAs, etc. in the materials and methods. You must also indicate in the methods the source, species, and catalog numbers/vendor identifiers (where appropriate) for all of your antibodies, including secondary. If

antibodies are not commercial, please add a reference citation if possible.

6) Microscope image acquisition: The following information must be provided about the acquisition and processing of images:

- a. Make and model of microscope
- b. Type, magnification, and numerical aperture of the objective lenses
- c. Temperature
- d. Imaging medium
- e. Fluorochromes
- f. Camera make and model
- g. Acquisition software
- h. Any software used for image processing subsequent to data acquisition. Please include details and types of operations involved (e.g., type of deconvolution, 3D reconstitutions, surface or volume rendering, gamma adjustments, etc.).

7) References: There is no limit to the number of references cited in a manuscript. References should be cited parenthetically in the text by author and year of publication. Abbreviate the names of journals according to PubMed.

8) Supplemental materials: There are strict limits on the allowable amount of supplemental data. Articles may have up to 5 supplemental figures and 10 videos. Please also note that tables, like figures, should be provided as individual, editable files. A summary of all supplemental material should appear at the end of the Materials and methods section. Please include one brief sentence per item.

9) Video legends: Should describe what is being shown, the cell type or tissue being viewed (including relevant cell treatments, concentration and duration, or transfection), the imaging method (e.g., time-lapse epifluorescence microscopy), what each color represents, how often frames were collected, the frames/second display rate, and the number of any figure that has related video stills or images.

10) eTOC summary: A ~40-50 word summary that describes the context and significance of the findings for a general readership should be included on the title page. The statement should be written in the present tense and refer to the work in the third person. It should begin with "First author name(s) et al..." to match our preferred style.

11) Conflict of interest statement: JCB requires inclusion of a statement in the acknowledgements regarding competing financial interests. If no competing financial interests exist, please include the following statement: "The authors declare no competing financial interests." If competing interests are declared, please follow your statement of these competing interests with the following statement: "The authors declare no further competing financial interests."

12) A separate author contribution section is required following the Acknowledgments in all research manuscripts. All authors should be mentioned and designated by their first and middle initials and full surnames. We encourage use of the CRediT nomenclature (<https://casrai.org/credit/>).

13) ORCID IDs: ORCID IDs are unique identifiers allowing researchers to create a record of their various scholarly contributions in a single place. At resubmission of your final files, please consider providing an ORCID ID for as many contributing authors as possible.

14) Materials and data sharing:

As a condition of publication, authors must make protocols and unique materials (including, but not limited to, cloned DNAs; antibodies; bacterial, animal, or plant cells; and viruses) described in our published articles freely available upon request by researchers, who may use them in their own laboratory only. All materials must be made available on request and without undue delay. We strongly encourage to deposit all the cell lines/strains and reagents generated in this study in public repositories. All datasets included in the manuscript must be available from the date of online publication, and the source code for all custom computational methods, apart from commercial software programs, must be made available either in a publicly available database or as supplemental materials hosted on the journal website. Numerous resources exist for data storage and sharing (see Data Deposition: <https://rupress.org/jcb/pages/data-deposition>), and you should choose the most appropriate venue based on your data type and/or community standard. If no appropriate specific database exists, please deposit your data to an appropriate publicly available database. Please, deposit your electron microscopy and mass spectrometry data in appropriate public databases.

15) Please note that JCB now requires authors to submit Source Data used to generate figures containing gels and Western blots with all revised manuscripts. This Source Data consists of fully uncropped and unprocessed images for each gel/blot displayed in the main and supplemental figures. Since your paper includes cropped gel and/or blot images, please be sure to provide one Source Data file for each figure that contains gels and/or blots along with your revised manuscript files. File names for Source Data figures should be alphanumeric without any spaces or special characters (i.e., SourceDataF#, where F# refers to the associated main figure number or SourceDataFS# for those associated with Supplementary figures). The lanes of the gels/blots should be labeled as they are in the associated figure, the place where cropping was applied should be marked (with a box), and molecular weight/size standards should be labeled wherever possible. Source Data files will be directly linked to

specific figures in the published article.

16) Journal of Cell Biology now requires a data availability statement for all research article submissions. These statements will be published in the article directly above the Acknowledgments. The statement should address all data underlying the research presented in the manuscript. Please visit the JCB instructions for authors for guidelines and examples of statements at (<https://rupress.org/jcb/pages/editorial-policies#data-availability-statement>).

B. FINAL FILES:

Thank you for this interesting contribution, we look forward to publishing your paper in Journal of Cell Biology.

Sincerely,

Melissa Gardner, PhD
Monitoring Editor
Journal of Cell Biology

Dan Simon, PhD
Scientific Editor
Journal of Cell Biology

Reviewer #1 (Comments to the Authors (Required)):

In this revised version the authors have addressed many of the concerns with the initial study. In particular, the demonstration that the P1 mutation in the motor domain of Kip2 indeed impairs tubulin binding in vitro, closes an important gap in the study. The manuscript now comprehensively describes two important Kip2 mutants: the deltaTail allele, defective in Bik1 binding and the P1 mutation in the motor domain, impaired in tubulin dimer, but not lattice binding. The combination of in vivo and in vitro

experiments allows the authors to propose an improved model for the mechanism of Kip2-mediated microtubule polymerization in yeast cells. The authors have also addressed most of my minor points in a convincing manner and I am therefore happy to support publication in the JCB.

Reviewer #2 (Comments to the Authors (Required)):

The authors have adequately addressed my comments and concerns in their revised manuscript.

Reviewer #3 (Comments to the Authors (Required)):

In general, the authors have done a nice job of addressing the reviewer concerns. I have a few concerns remaining that I feel can be addressed with text revisions, but also feel it is important they are addressed prior to publication. Several of my comments are in the vein of one of Reviewer 2's prior concern, in that statements about binding rates and dwell times of Kip2, which were inferred from varying parameters in simulations are conflated with actual observations and stated/discussed without proper qualification.

1) The modeling results in the results and discussion sections are presented as definitively disclosing what is occurring *in vivo*. This aspect is significantly overstated. At best, the modeling results are consistent with some possible scenario, or suggestive of what might be occurring *in vivo*. These instances of modeling results must be framed in the proper context. As an example, the top of page 9 states: "In addition to this, the modelling results showed that in wild type cells and both mutants, Kip2 is much more likely to start a run along the microtubule at its minus end than at any other position on the microtubule: Kip2's on-rate constants were orders of magnitude lower than its in-rate constants, as previously observed (37). Thus, disrupting the Bik1-Kip2 interaction did not affect dramatically Kip2 recruitment at the microtubule minus-end, i.e., the SPB. The estimates, however, indicated as well that Kip2's on-rate and in-rate constants were both significantly increased in *bik1Δ* and to a lesser extent in the Kip2-ΔT-3xsfGFP mutant cells (Fig. 3G)...". The modelling did not show what Kip2 does in cells. It can show what Kip2 'may' do in cells, and real experimental evidence is needed to support the possibility. This is not made clear in sections of the results and discussion.

2) Page 14 the discussion states: "The second is a binding site for the cytoplasmic linker protein Bik1 that increases Kip2 residence time at microtubule plus-ends in living cells, in a Bik1-dependent manner." It should be clear what aspects of this conclusion/discussion are supported by results obtained *in vivo/vitro*, and which are inferred from varying simulation parameters.

3) Similarly in the second paragraph on page 15 statements that are based on simulations are convoluted with statements based on observations *in vivo* and *in vitro*. The basis for such statements should be clear when they are based on output generated by simulated scenarios. For example: "Both the loss of this interaction or Bik1 altogether decreases the time Kip2 spends at microtubule plus-ends."

4) Another example from the middle of page 15: "Thus, for Kip2 to promote microtubule growth, it needs to linger around the microtubule plus-end for some time, in a Bik1-dependent manner." This statement as a conclusion may be OK, as long as the previous inferences that are based on simulations with altered parameters are clearly noted as described above. Otherwise, a statement like this should be qualified such as "We propose" or "Our modeling results indicate Kip2 likely needs to linger..."

5) There may be more instances as those illustrated above. They only serve as examples of where the concern occurs.

6) Page 14: "We suggest that once arrived at the very end of a microtubule, through its motile translocation along the shaft, the free motor domain of Kip2 binds free tubulin and promotes its incorporation in the protofilament. Thus, Kip2 might elongate its own track under its own "feet" once arrived at microtubule tips." The authors may already intend this to be the case, but this description makes it sound like Kip2 is exclusively binding, or capturing, free tubulin and then bringing it into the microtubule. While it's fine to suggest this based on the data, a situation in which Kip2 at the microtubule end interacts with the newly added tubulin dimers and stabilizes their incorporation. It should be noted that this process is not excluded by the data.

7) Middle of page 15: "This result and the fact that Kip2 needs to dwell at the microtubule plus-end...". When reading the few sentences preceding this one, it is unclear exactly what "This result" refers to. It may refer to the same result, i.e. that Kip2 dwells at the tip. Perhaps this sentence could be clarified.

Zürich, March 10, 2023

Dear Editors,

It is with great happiness that we have learned about your decision to publish our manuscript "The motor domain of the kinesin Kip2 promotes microtubule polymerization at microtubule tips" in the Journal of Cell Biology. We have now gone through your final requests and edited the text and figures accordingly. A response in which we explain how we adapted the text to satisfy the last concerns of reviewer #3 can be found below. We thank you very much for publishing our paper, which we are very much looking forwards to see in print.

With my best regards,

Yves Barral

Response to reviewer #3:

In general, the authors have done a nice job of addressing the reviewer concerns. I have a few concerns remaining that I feel can be addressed with text revisions, but also feel it is important they are addressed prior to publication. Several of my comments are in the vein of one of Reviewer 2's prior concern, in that statements about binding rates and dwell times of Kip2, which were inferred from varying parameters in simulations are conflated with actual observations and stated/discussed without proper qualification.

Regarding the general role of modeling in this work, we wish to clarify (see below for details) that we used the model primarily as an analytic tool for estimating parameter values from *in vivo* observations. This function of our model is identical to the case where we use observations such as spatial positions of microtubule tips at discrete time points and a linear regression model to estimate parameters that are not directly observable, such as microtubule growth and shrinkage rates. For our model based on ordinary differential equations, this estimation involves simulations because a closed-form solution as in linear regression is not possible, and by multiple simulations in a Bayesian approach, we obtain both the parameter estimates and their uncertainties. Our model itself was previously validated (ref) with *in vivo* data and it is based on physical constraints (mass balances), chemical kinetics, and measured parameters (e.g., motor speeds) whenever possible, making it suitable for the inference we are performing here. Finally, we only used the model for prediction instead of inference in one case, when evaluating the predicted impact of *in vitro* parameters on Kip2 distributions (bottom of page 8).

1) The modeling results in the results and discussion sections are presented as definitively disclosing what is occurring *in vivo*. This aspect is significantly overstated. At best, the modeling results are consistent with some possible scenario, or suggestive of what might be occurring *in vivo*. These instances of modeling results must be framed in the proper context. As an example, the top of page 9 states: "In addition to this, the modelling results showed that in wild type cells and both mutants, Kip2 is much more likely to start a run along the

microtubule at its minus end than at any other position on the microtubule: Kip2's on-rate constants were orders of magnitude lower than its in-rate constants, as previously observed (37). Thus, disrupting the Bik1-Kip2 interaction did not affect dramatically Kip2 recruitment at the microtubule minus-end, i.e., the SPB. The estimates, however, indicated as well that Kip2's on-rate and in-rate constants were both significantly increased in *bik1*Δ and to a lesser extent in the Kip2-ΔT-3xsfGFP mutant cells (Fig. 3G)...". The modelling did not show what Kip2 does in cells. It can show what Kip2 'may' do in cells, and real experimental evidence is needed to support the possibility. This is not made clear in sections of the results and discussion.

We have revised the text (p.8, 1st para) to specify the role of the model with the rationale above. To distinguish inferred results from direct observations, the text now uses 'inferred parameter values' (i.e. values estimated using the dynamic model and *in vivo* observations) consistently. For example, the sentence quoted by the reviewer now reads: "In addition to this, our estimates for Kip2's on-rate constants were orders of magnitude lower than its in-rate constants, as previously observed (37). Thus, in wild type cells and both mutants, Kip2 is much more likely to start a run along the microtubule at its minus-end than at any other position on the microtubule." Regarding 'real experimental evidence', we refer to the rationale above – parameters such as on-rate constants are not directly observable, neither *in vivo* nor *in vitro*.

2) Page 14 the discussion states: "The second is a binding site for the cytoplasmic linker protein Bik1 that increases Kip2 residence time at microtubule plus-ends in living cells, in a Bik1-dependent manner." It should be clear what aspects of this conclusion/discussion are supported by results obtained *in vivo/vitro*, and which are inferred from varying simulation parameters.

We have re-phrased the sentence to: "The second is a binding site for the cytoplasmic linker protein Bik1 that, according to our estimates, seems to increase Kip2 residence time at microtubule plus-ends in living cells, in a Bik1-dependent manner." With the consistent use of "estimates", we hope that the context is now clear.

3) Similarly in the second paragraph on page 15 statements that are based on simulations are convoluted with statements based on observations *in vivo* and *in vitro*. The basis for such statements should be clear when they are based on output generated by simulated scenarios. For example: "Both the loss of this interaction or Bik1 altogether decreases the time Kip2 spends at microtubule plus-ends."

The basis for this statement was given in the next sentence (now combined as one sentence to clarify this basis): "... our out-rate constant estimates for Kip2 dissociation are consistent with Kip2 dissociating as soon as it arrives at the microtubule plus-end when Bik1 is absent."

4) Another example from the middle of page 15: "Thus, for Kip2 to promote microtubule growth, it needs to linger around the microtubule plus-end for some time, in a Bik1-dependent manner." This statement as a conclusion may be OK, as long as the previous inferences that are based on simulations with altered parameters are clearly noted as described above. Otherwise, a statement like this should be qualified such as "We propose" or "Our modeling results indicate Kip2 likely needs to linger..."

We have now inserted a reference to the figure containing the experimental observations.

5) There may be more instances as those illustrated above. They only serve as examples of where the concern occurs.

We went through the entire text to address the concern of the reviewer throughout it.

6) Page 14: "We suggest that once arrived at the very end of a microtubule, through its motile translocation along the shaft, the free motor domain of Kip2 binds free tubulin and promotes its incorporation in the protofilament. Thus, Kip2 might elongate its own track under its own "feet" once arrived at microtubule tips." The authors may already intend this to be the case, but this description makes it sound like Kip2 is exclusively binding, or capturing, free tubulin and then bringing it into the microtubule. While it's fine to suggest this based on the data, a situation in which Kip2 at the microtubule end interacts with the newly added tubulin dimers and stabilizes their incorporation. It should be noted that this process is not excluded by the data.

We are now acknowledging the possibility that Kip2 stabilizes the newly added tubulin dimer. The sentence now reads "Whereas Kip2 might simply stabilize newly incorporated tubulin at the microtubule end, we suggest that once arrived at the very end of a microtubule, through its motile translocation along the shaft, the free motor domain of Kip2 actually binds free tubulin and promotes its incorporation in the protofilament.»

7) Middle of page 15: "This result and the fact that Kip2 needs to dwell at the microtubule plus-end...". When reading the few sentences preceding this one, it is unclear exactly what "This result" refers to. It may refer to the same result, i.e. that Kip2 dwells at the tip. Perhaps this sentence could be clarified.

Indeed, the repetition of the concept of dwelling (lingering in the previous sentence) is confusing. We have therefore deleted the phrase "and the fact that Kip2 needs to dwell at the microtubule plus-end".